# EMERGE: A Benchmark for Updating Knowledge Graphs with Emerging Textual Knowledge

## Abstract

Knowledge Graphs (KGs) are structured knowledge repositories containing entities and relations between them. In this paper, we study the problem of automatically updating KGs over time in response to evolving knowledge in unstructured textual sources. Addressing this problem requires identifying a wide range of update operations based on the state of an existing KG at a given time and the information extracted from text. This contrasts with traditional information extraction pipelines, which extract knowledge from text independently of the current state of a KG. To address this challenge, we propose a method for construction of a dataset consisting of Wikidata KG snapshots over time and Wikipedia passages paired with the corresponding edit operations that they induce in a particular KG snapshot. We obtain these pairs by aligning annotated hyperlinked entity mentions in each Wikipedia passage with the corresponding entities involved in the updated Wikidata triples. We verify, using LLMs with human validation, that these textual passages contain the knowledge needed to support the associated KG edits. The resulting dataset comprises 233K Wikipedia passages aligned with a total of 1.45 million KG edits over 7 different yearly snapshots of Wikidata from 2019 to 2025. Our experimental results highlight key challenges in updating KG snapshots based on emerging textual knowledge, particularly in integrating knowledge expressed in text with the existing KG structure. These findings position the dataset as a valuable benchmark for future research. We will publicly release our dataset and model implementations.[1]

## 1 Introduction

Knowledge graphs (KGs) play a crucial role in applications such as question answering (Wang et al., 2024; Dong et al., 2025), recommender systems (Zhang et al., 2024; Wang et al., 2025), information retrieval (Reinanda et al., 2020), fact-checking (Kim et al., 2023; Hao & Wu, 2025), and healthcare prediction (Jiang et al., 2025), among others (Zou, 2020). Furthermore, KGs provide structured, queryable world knowledge that increasingly complements *large language models (LLMs)* (Pan et al., 2024; Cai et al., 2025). This integration has been used to reduce LLM hallucinations (Agrawal et al., 2024; Lavrinovics et al., 2025), improve fine-tuning (Chen et al., 2025; Ma et al., 2025), enhance planning (Chen et al., 2024; Petruzzellis et al., 2025), support complex reasoning (Sun et al., 2024; Luo et al., 2025), and provide reliable knowledge augmentation (Han et al., 2024; Li et al., 2025c). However, as world knowledge evolves, KGs must also be updated to remain reliable (Polleres et al., 2023; Hofer et al., 2024; Li et al., 2025b). Yet, existing temporal KG benchmarks (Liang et al., 2024; Alam et al., 2024) model only internal temporal dynamics and do not capture how KGs should be updated in response to new world knowledge emerging in external textual sources. In addition, current textual information extraction (IE) datasets and models (Zhao et al., 2024b; Xu et al., 2024) do not link extracted facts to the concrete KG updates they should induce.

To address these limitations, we introduce EMERGE, a novel, automatically constructed benchmark that aligns emerging textual knowledge with the concrete updates it induces in a KG. Concretely, EMERGE links evolving changes in the Wikidata KG (Vrandečić & Krötzsch, 2014) with

---

[1]Code and dataset will be released upon acceptance. The test set is included in the supplementary material.

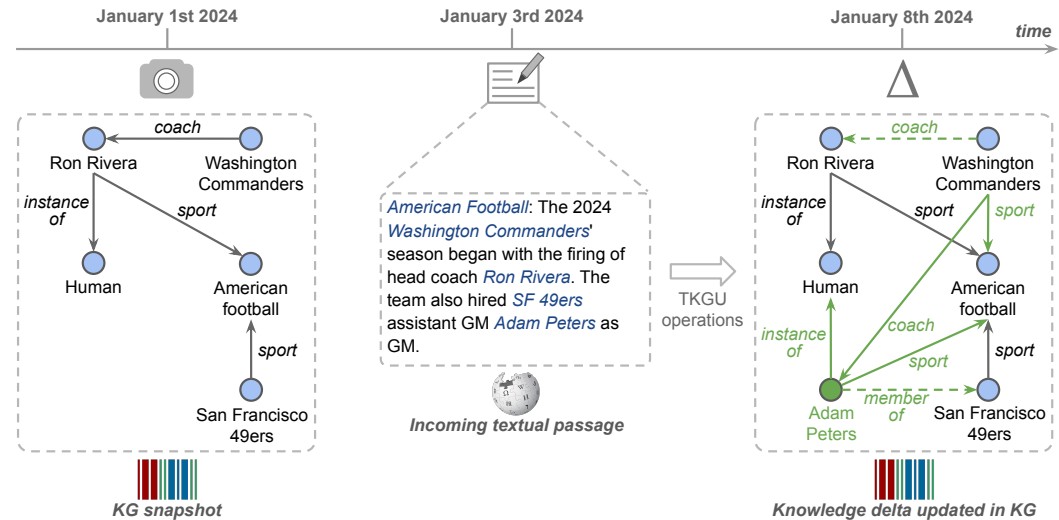

Figure 1: Illustration of one instance in EMERGE. The reference KG *snapshot* of January 1st 2024 is updated with new, *emerging knowledge* contained in the incoming *textual passage* from January 3rd 2024. The *updated KG* involves not only creation of new relations (solid green arrows), but also generation of new entities (green circle) and deprecation of relations (dashed green arrows).

corresponding textual passages from Wikipedia that reflect these KG updates over time. This alignment enables evaluating both (i) how well models integrate new textual knowledge into a KG and (ii) how temporally evolving KG structures (i.e., KG schema) affect this integration. Furthermore, EMERGE is incrementally extensible through an automatic pipeline that continuously incorporates new knowledge from Wikipedia and Wikidata. Figure 1 illustrates an example in which a *KG snapshot* from January 1st, 2024 is updated based on *emerging textual evidence* from January 3rd, 2024. To construct such instances, EMERGE aligns weekly *knowledge deltas* in Wikidata with the corresponding textual changes in Wikipedia.

Furthermore, EMERGE differs from existing mainstream IE datasets, which primarily focus on extracting triples from text, either via manual annotations or by linking text to a static KG. As a result, these existing benchmarks cannot evaluate the broader set of operations needed to update a KG to reflect textual knowledge. Such operations require creation new entities, linking them to existing ones, and deprecation of outdated facts (see Figure 1 for an example). To capture these requirements, EMERGE defines five text-driven KG updating (TKGU) operations (see Section 3) and is, to our knowledge, the first dataset to support all of them (see a detailed comparison with existing benchmarks in Appendix A.1). Our benchmarking further shows that state-of-the-art IE models fall short in supporting the full range of operations required to update KGs with new knowledge (see Table 1). Furthermore, these models rely solely on knowledge expressed in text and remain unaware of how that knowledge is structured within a KG. As a result, the extracted triples, though semantically valid, often fail to align with the KG schema and structure.

In summary, the contributions of this paper are as follows:

- We formalize and study the problem of maintaining KGs from emerging textual knowledge, defining it through a set of fundamental text-driven KG updating (TKGU) operations.

- EMERGE, a novel dataset that maps emerging knowledge in textual passages to corresponding updates in temporally evolving KG snapshots.

- A publicly available pipeline for extending EMERGE with new KG snapshots, enabling the evaluation of models on continuously evolving knowledge.

- Experimental results and analysis on EMERGE using two state-of-the-art IE architectures.

Table 1: Comparison of state-of-the-art information extraction models by the type of extracted knowledge: (1) existing KG triples (*X-Triples*), (2) new triples with existing KG entities (*E-Triples*), (3) new triples with emerging entities (*EE-Triples*), (4) new triples linking emerging entities to the rest of the KG (*EE-KG-Triples*), and (5) deprecated triples (*D-Triples*). The *KG Link* column indicates whether extracted triples are linked to a KG.

| Model | KG Link | Supported textual knowledge type extraction | | | | |
| --- | --- | --- | --- | --- | --- | --- |
| | | *X-Triples* | *E-Triples* | *EE-Triples* | *EE-KG-Triples* | *D-Triples* |
| REBEL (2021) | ✗ | ✓ | ✓ | ✓ | ✗ | ✗ |
| GenIE (2022) | ✓ | ✓ | ✓ | ✗ | ✗ | ✗ |
| KnowGL (2023) | ✓ | ✓ | ✓ | ✓ | ✗ | ✗ |
| GCD (2023) | ✓ | ✓ | ✓ | ✗ | ✗ | ✗ |
| ReLiK cIE (2024) | ✓ | ✓ | ✓ | ✗ | ✗ | ✗ |
| ReLiK RE (2024) | ✗ | ✓ | ✓ | ✓ | ✗ | ✗ |
| EDC (2024) | ✗ | ✓ | ✓ | ✓ | ✗ | ✗ |
| ATG (2024) | ✗ | ✓ | ✓ | ✓ | ✗ | ✗ |
| CodeKGC (2024) | ✗ | ✓ | ✓ | ✓ | ✗ | ✗ |

## 2 RELATED WORK

Below we describe the most relevant related work directions. Additionally, we provide an extensive related work section and a comparison table (Table 4) in Appendix A.

**KG completion and refinement.** Research on KG completion (KGC) (Shen et al., 2022) and refinement (Paulheim, 2016; Subagdja et al., 2024) has produced many datasets aimed at predicting missing relations between entities. Early work introduced WN18 and FB15k Bordes et al. (2013), derived from WordNet (Miller, 1995) and Freebase (Bollacker et al., 2008), followed by improved variants such as WN18RR and FB15k-237 (Toutanova & Chen, 2015; Dettmers et al., 2018) addressing redundancy and data leakage. Larger and more recent datasets include Wikidata5M Wang et al. (2021), along with Wiki/NELL-One (Xiong et al., 2018), FB15K-237N (Lv et al., 2022), CoDEx (Safavi & Koutra, 2020), YAGO3-10 (Mahdisoltani et al., 2014), and LiterallyWikidata (Gesese et al., 2021). While these datasets evaluate models on predicting new edges within the KG, they remain restricted to the KG internal structure. Our objective instead is to support KG updates driven by the information originating in external unstructured textual sources. This distinction also separates our work from temporal KG completion (TKGC) datasets such as GDELT (Leetaru & Schrodt, 2013), ICEWS14/05-15 (Garcia-Duran et al., 2018), Wikidata12k (Dasgupta et al., 2018), Wikidata-big (Lacroix et al., 2020), ICEWS18 (Jin et al., 2020), and more recently TGB and TGB 2.0 (Huang et al., 2024; Gastinger et al., 2024), among others (Liang et al., 2024; Alam et al., 2024). While these benchmarks capture internal temporal evolution of facts, they do not model how KGs should be kept up to date with knowledge emerging in external textual sources. EMERGE fills this gap by aligning such external textual evidence with the concrete KG updates it induces, enabling the study of models that keep KGs updated as world knowledge evolves.

**Information extraction (IE).** To evaluate the ability of models to extract structured knowledge, researchers have developed IE datasets by annotating entity relations. MUC-7 (Chinchor & Marsh, 1998) introduced three relation types, with later datasets expanding in size, relation diversity, or both. Notable examples include CoNLL04 (Roth & Yih, 2004), ACE 2005 (Walker et al., 2006), ERE (Aguilar et al., 2014; Song et al., 2015), BC5CDR (Li et al., 2016), TACRED (Zhang et al., 2017), SciERC (Luan et al., 2018), SemEval-2010 (Hendrickx et al., 2010), SemEval-2017 (Augenstein et al., 2017), DWIE (Zaporojets et al., 2021) and BioRED (Luo et al., 2022), among others. Other datasets, such as NYT (Riedel et al., 2010), explicitly linked KG triples to textual snippets using distant supervision. Similarly, but on a larger scale, Gabrilovich et al. (2013) introduced FACC1 by aligning ClueWeb12 documents with Freebase entity mention annotations. In parallel, the TAC-KBP challenges (Ji et al., 2010; TAC-KBP, 2022) (2009 – 2020) produced proprietary manually annotated datasets for knowledge base population tasks such as slot filling and entity linking. More recently, these resources have been extended with a variety of datasets that map textual knowledge to KG triples, such as WebNLG (Gardent et al., 2017), KELM (Agarwal et al., 2021), FewRel

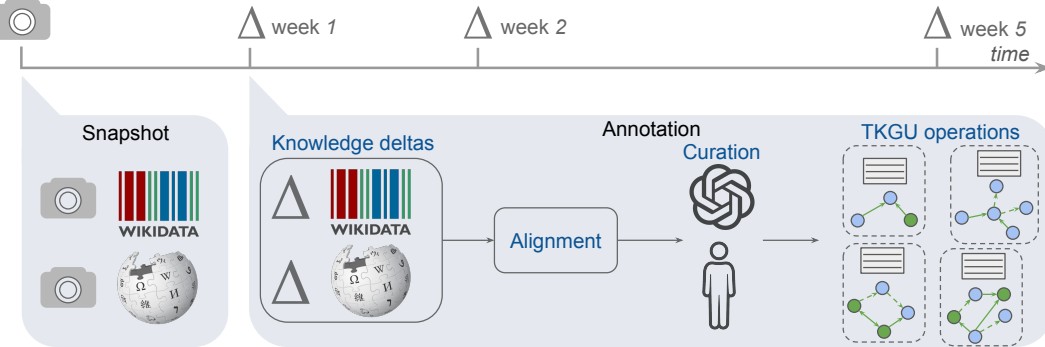

Figure 2: Illustration of EMERGE creation pipeline. First, weekly *knowledge deltas* ($\Delta$) are extracted by identifying changes in Wikipedia passages and Wikidata KG relative to a fixed *snapshot*. In the *Alignment* step, these KG and textual deltas are connected. During *Curation*, an LLM discards KG updates not supported by aligned textual changes, a process verified with manual annotations on a subsample of alignments. The result is high-quality text–KG update pairs, as in Figure 1, where multiple *TKGU operations* (Section 3) update the KG with emerging textual knowledge.

(Han et al., 2018), DocRED (Yao et al., 2019), Wiki/GEO-NRE (Distiawan et al., 2019), BioRel (Xing et al., 2020), T-REX (Elsahar et al., 2019) and REBEL (Cabot & Navigli, 2021). While these datasets connect textual knowledge to KG triples and literals, they do not account for the operations required to update a KG as new information emerges in text. Our work addresses this gap by linking new textual knowledge to the specific update operations (see Section 3) on a KG snapshot. Methodologically, existing state-of-the-art IE methods (see Table 1) provide a natural starting point for tackling TKGU operations, as they extract structured knowledge from text and already cover some of these operations. However, they remain largely oblivious to the existing KG structure and require extensions to integrate emerging textual content into the KG, such as deprecating outdated triples, adding new entities, and enforcing structural consistency based on how entities and relations are used within the KG.

## 3 PROBLEM DEFINITION

We define the problem of *text-driven knowledge graph updating* (TKGU) as determining the necessary edits to a KG at a particular point in time, given a textual passage. More formally, we define a KG *snapshot* at some point time $t$ as a tuple $G_t = (V_t, R_t, T_t)$ where $V_t$ is a set of entities, $R_t$ is a set of relation types, and $T_t$ is a set of *triples* of the form $(s, p, o)$ where $s, o \in V_t$ are the subject and object, and $p \in R_t$ is the relation between them. Given a textual passage $d_{t'}$ created at some point in time $t' > t$, the task consists in generating a set of TKGU operations defined as follows:

**Emerging triples (E-Triples).** Addition of triples that are not present in the KG but involve entities that already exist in it; that is, $(s, p, o) \notin G_t$ and and $s \in V_t \wedge o \in V_t$. For example, in Figure 1, the added triple *(Washington Commanders, sport, American Football)* involves the entities *Washington Commanders* and *American Football*, both of which already exist in the KG.

**Emerging entities and triples (EE-Triples).** Addition of triples that do not exist in the KG and involve a subject entity, object entity, or both that are also absent. That is, $(s, p, o) \notin G_t$ and $s \notin V_t \vee o \notin V_t$. For example, in Figure 1, the added triple *(Washington Commanders, coach, Adam Peters)* introduces the entity *Adam Peters*, which is not yet in the KG.

**Emerging entities to KG triples (EE-KG-Triples).** Addition of new triples where exactly one of the subject or object entities is mentioned in a passage $d_{t'}$, while the other already exists in the KG and is not explicitly mentioned in the passage. These triples evaluate the ability of the models to integrate newly emerging entities by linking them to existing ones in the KG. For example, in Figure 1, the added triple *(Adam Peters, instance of, Human)* links the emerging entity *Adam Peters* to the existing entity *Human*, even though this relation is not explicitly stated in the passage.

**Deprecated triples (D-Triples).** Deprecation of triples already existing in a KG based on emerging evidence in textual passage. For example, in Figure 1, the triples *(Adam Peters, member of, San Francisco 49ers)* and *(Washington Commanders, coach, Ron Rivera)* are deprecated based on updated information in the passage.

**Existing triples (X-Triples).** Detection of triples already existing in the KG that are supported by textual passage, i.e., $(s, p, o) \in G_t$. This operation evaluates the ability of models to recognize existing knowledge. For example, in Figure 1, the triple *(San Francisco 49ers, sport, American football)* is both supported by the passage and already present in the original KG snapshot.

Table 1 compares existing IE architectures based on the types of TKGU operations (see above) they are able to extract. While many models can extract triples involving existing entities in a KG (*X-Triples* and *E-Triples*), most struggle to identify triples with emerging entities (*EE-Triples*) and none of them supports linking them to the rest of the KG (*EE-KG-Triples*). Furthermore, some methods only partially integrate newly extracted knowledge, as they do not link the extracted triples to the KG (see *KG* column). For example, *relation extraction* models such as REBEL (Cabot & Navigli, 2021) and ReLiK RE (Orlando et al., 2024) are able to extract new triples but do not link their entities and relations to the KG; other models such as EDC (Zhang & Soh, 2024), link only relations but not entities. Finally, existing IE methods, to the best of our knowledge, are not designed to identify triples that should be deprecated based on emerging textual knowledge (*D-Triples*).

## 4 OUR DATASET

We introduce EMERGE, a large-scale dataset that, unlike existing benchmarks, supports all the TKGU operations defined in Section 3.

### 4.1 DATA COLLECTION

We construct a dataset consisting of 7 Wikidata yearly snapshots taken on January 1st at 00:00 GMT from 2019 to 2025. We expect that these snapshots will enable to evaluate the drift in temporal performance of models pre-trained at different time points. To evaluate the ability of the models to update KG with emerging knowledge, we generate cumulative weekly *deltas* (up to 5 weeks) for each snapshot (see Figure 2). Each delta represents a time window and includes textual passages along with the corresponding KG updates occurring during that period. Below, we describe in more detail the main steps in the EMERGE dataset creation pipeline.

**Wikipedia and Wikidata dumps.** We begin by downloading the historical revision logs from the Wikipedia and Wikidata dumps available at `https://dumps.wikimedia.org/`. These logs provide complete access to the revision history of Wikipedia and Wikidata, enabling fine-grained tracking of temporal changes. Using this level of granularity, we are able to construct EMERGE using *any number of arbitrarily defined KG snapshots and delta windows, with temporal precision down to the second*. This capability sets EMERGE apart from existing datasets designed to evaluate model performance on evolving KG knowledge (Boschee et al., 2015; Dasgupta et al., 2018; Lacroix et al., 2020), which are typically derived from a single KG snapshot and rely only on temporal attributes associated with edges. While such datasets are valuable for predicting the emergence of new facts over time, they do not allow the evaluation of how structural changes in the KG across different snapshots affect model performance. Moreover, because we have access to the full revision history of Wikipedia pages, we can evaluate models on all the newly introduced textual content within any chosen temporal delta. This allows us to assess, for instance, how varying the size of delta windows influences model performance. It also contrasts with related datasets using Wikipedia (Lewis et al., 2020; Jang et al., 2022a; Onoe et al., 2023; Zhao et al., 2024a), which are based on only one or a small number of manually downloaded Wikipedia snapshots, thereby limiting temporal flexibility.

**Snapshot generation.** Given a list of desired snapshot timestamps, we process Wikipedia and Wikidata history revisions to obtain the following components for each timestamp $t$: (1) a Wikidata KG snapshot $G_t$ corresponding to $t$, (2) a dictionary of entities present in Wikipedia at $t$, along with their corresponding textual descriptions, and (3) a dictionary of relation types present in Wikidata at $t$ with definitions. In line with the Wikidata5M dataset (Wang et al., 2021), we restrict the Wikidata KG to include only entities that are present in Wikipedia.

**KG deltas generation.** For each snapshot, we generate deltas in weekly increments, spanning up to 5 weeks. Each delta represents the difference between two KG snapshots, denoted as $G_{t+\Delta} - G_t$, where $\Delta$ represents the delta window. Each of the resulting deltas involve KG triple operations outlined in Section 3. Concretely, *X-Triples* exist in $G_t$ and $G_{t+\Delta}$, *E-Triples* contain new emerging relations in $G_{t+\Delta}$ between entities already existing in $G_t$, and *EE-Triples* and *EE-KG-Triples* consist of emerging relations between entities where subject or object do not exist in $G_t$, and is introduced in $G_{t+\Delta}$. Finally, to obtain *D-Triples*, besides including the removed edges, we match Wikidata triple qualifiers (see Appendix J) that explicitly indicate knowledge deprecation within the delta interval. We mark these triples as deprecated rather than removing them, since the underlying fact does not change but expires within the delta interval.

**Aligning KG deltas with text.** For each delta in a given snapshot $t$, we retrieve the newly introduced Wikipedia passages within the temporal window corresponding to that delta. Following the approach of Cabot & Navigli (2021); Elsahar et al. (2019), we then *align* these passages with triples in each of the KG deltas by matching the annotated hyperlinked entity mentions in each of the passages to the corresponding entities in the triples. We refer to this distant supervision process as the *alignment* step (see Figure 2). The resulting text-triple pairs are subsequently refined in the *curation* step (see Section 4.2) to retain only those pairs in which the textual content supports the associated TKGU operations defined in Section 3.

## 4.2 Quality Assurance and Control

During the *alignment* step of EMERGE creation pipeline (see Figure 2) we use multiple heuristics to ensure the quality of the aligned textual passages with KG updates. For instance, we filter out passages with a low proportion of English words and those containing wikitext special symbols used for constructing elements such as tables and images. Furthermore, we discard updates in Wikidata and Wikipedia that are quickly rolled back, as these often indicate incorrect or vandalized changes. A complete list of preprocessing and cleaning steps can be found in Section L.4 in the appendix.

During the *Curation* step of the EMERGE pipeline (see Figure 2), we use `Meta-Llama-3.1-405B` to validate that all TKGU operations can be derived from the corresponding textual passage. The full prompt design and illustrative examples are provided in Appendix C.1. This step flags KG updates not supported by the text, rather than removing them, enabling future use of more powerful LLMs for additional verification and curation. Preserving unsupported triples also allows evaluation of potential models that may rely less on text and more on KG knowledge, particularly for EE-KG-Triples TKGU operations, where an entity may not appear in the passage and updating the KG requires KG knowledge itself (e.g., all humans in the KG link to the entity *human*). Appendix C.3 reports additional statistics on the fraction of triples marked as unsupported.

Finally, during the *Curation* step, we manually annotate a random subset of 500 triple-text pairs (100 per TKGU operation type) to verify agreement with the LLM. We observe *Strong* to *Almost perfect* agreement depending on the operation type, supporting the use of `Meta-Llama-3.1-405B` to annotate the full dataset. Detailed annotation guidelines and agreement statistics are provided in Appendices C.2.1 and C.2.2, respectively.

## 4.3 Dataset Statistics

EMERGE consists of 233K instances across seven yearly KG snapshots (2019–2025), with a total of 1.45M TKGU update operations. Updates in each snapshot are evaluated over cumulative weekly delta ($\Delta$) intervals of up to 5 weeks. Both the KG size (i.e., number of entities and edges) and the schema (i.e., number of relation types) evolve across snapshots. For instance, the 2019 KG snapshot contains 5.96M entities, 25.73M relations, and 5,646 relation types, while the 2025 snapshot includes 6.93M entities, 37.54M relations, and 12,304 relation types. This dynamic setting enables the evaluation of model robustness under evolving KG knowledge and schema changes, thereby *reflecting real-world KG evolution*. Additional tables and figures in Appendix D provide a detailed overview of the size and distribution of TKGU operations in EMERGE. Furthermore, Tables 7–12 in Appendix E present illustrative examples of each TKGU operation type introduced in Section 3.

## 4.4 DATASET EXTENSION

EMERGE is an automatically constructed dataset, which we plan to extend using yearly snapshots of Wikipedia and Wikidata, following the pipeline described in Section 4 and illustrated in Figure 2. These periodic extensions will enable the evaluation of architectures on their ability to extract emerging real-world knowledge from text. This is particularly important for LLM-based architectures, which are prone to hallucinating outdated information due to their internal parameters being pre-trained on older textual sources (Wu et al., 2024a). To facilitate further development, we will also provide code that allows users to extend the dataset themselves.

## 5 EXPERIMENTAL SETUP

We evaluate EMERGE using two state-of-the-art information extraction (IE) models that extract structured knowledge as triples from text. These models are tested on a set constructed by subsampling 5,000 instances from each snapshot (1,000 per delta), resulting in a total of 35,000 instances and 201,369 TKGU operations. During subsampling, we retained up to 400 instances per delta containing D-Triples TKGU operations. This ensures a sufficiently large number of D-Triples examples for evaluation, even though they account for only 0.6% of all TKGU operations in the full dataset. Conversely, in the test set, D-Triples constitute 3.3% (6,718 operations) of all TKGU operations. This low proportion of D-Triples does not affect metric stability, as each TKGU operation type is evaluated independently rather than through aggregated performance across types (see Table 2). A detailed comparison of TKGU operation distributions is provided in Appendix D.2.

### 5.1 MODELS

To assess state-of-the-art performance on EMERGE, we evaluate two widely used IE architectures: traditional extractive span-based models (Lee et al., 2017) and recent generative large language models (LLMs) (Dagdelen et al., 2024; Xu et al., 2024; Zhang et al., 2025). For the span-based setting, we use ReLiK (Orlando et al., 2024), and for the LLM-based setting, we adopt EDC (Zhang & Soh, 2024). Rather than comparing these models in terms of absolute performance, our goal is to illustrate the complementary limitations of two mainstream IE paradigms when applied to text-driven KG updating. This setup highlights where each paradigm succeeds or fails across the different TKGU operations defined in Section 3, particularly in their ability to handle emerging entities and reason over existing KG structure. Below, we describe these architectures in more detail and explain how we adapt them to each TKGU operation type.

**ReLiK.** ReLiK (Orlando et al., 2024) is a highly scalable architecture designed to minimize resource usage while achieving state-of-the-art performance in both entity linking and relation extraction. In our study, we evaluate two variants of ReLiK: closed information extraction ReLiK (ReLiK cIE) and relation-extraction ReLiK (ReLiK RE). *ReLiK cIE* operates under the closed IE assumption (Galárraga et al., 2014; Chaganty et al., 2017; Josifoski et al., 2023), predicting relations only between entities already present in the KG. Consequently, it can handle only those TKGU operations involving known entities, namely, *X-Triples* and *E-Triples* as defined in Section 3. For each test snapshot $t$, both models are provided with the corresponding KG snapshot. Specifically, ReLiK cIE receives the dictionaries of entities ($V_t$) and relation types ($R_t$) present in $t$, while ReLiK RE is given only the relation types ($R_t$), as it predicts relations without linking extracted entity mentions. Further details on the ReLiK execution and configuration are provided in Appendix I.

**EDC.** The *extract, define, canonicalize (EDC)* framework, introduced by Zhang & Soh (2024), is a state-of-the-art LLM-based approach. We adapt the original EDC prompt to additionally extract triples involving entities that are not explicitly mentioned in the input text but are potentially present in a Wikidata KG snapshot. Furthermore, we extend this prompt even further, asking the model to identify potential triples to be deprecated from the KG. This way, we give the model the ability to identify *EE-KG-Triples* and *D-Triples* operations based on the emerging evidence in text (see Section 3). We term this adaptation **EDC+** in our experiments, and evaluate it on `Mistral-7B-Instruct-v0.2` (*EDC+ Mistral-7b*) and `gemma-7b` (*EDC+ Gemma-7b*) LLMs. Additional execution details as well as the used prompts are described in Appendix H.

Table 2: *Recall* (measured using the completeness score) for IE models that do not link extracted triples to the KG, evaluated across KG snapshots on the TKGU operations defined in Section 3.

| TKGU | Model | 2019 | 2020 | 2021 | 2022 | 2023 | 2024 | 2025 |
|------|-------|------|------|------|------|------|------|------|
| X-Triples | EDC+ Mistral-7b | 9.7 | 7.5 | 10.5 | 8.1 | 11.7 | 7.4 | 8.7 |
| | EDC+ Gemma-7b | 7.5 | 7.9 | 7.6 | 5.8 | 8.1 | 5.7 | 6.5 |
| | ReLiK RE | 25.3 | 24.5 | 24.1 | 20.1 | 22.1 | 19.2 | 20.3 |
| E-Triples | EDC+ Mistral-7b | 18.8 | 17.6 | 16.3 | 17.1 | 18.6 | 19.4 | 19.3 |
| | EDC+ Gemma-7b | 16.4 | 14.4 | 13.0 | 13.5 | 15.7 | 14.5 | 14.6 |
| | ReLiK RE | 23.3 | 20.3 | 23.1 | 15.9 | 17.0 | 15.0 | 16.4 |
| EE-Triples | EDC+ Mistral-7b | 21.3 | 16.7 | 10.0 | 15.7 | 18.4 | 13.2 | 15.6 |
| | EDC+ Gemma-7b | 18.4 | 13.5 | 9.1 | 14.8 | 17.2 | 13.0 | 13.2 |
| | ReLiK RE | 25.4 | 18.7 | 12.4 | 23.7 | 22.4 | 15.6 | 16.2 |
| EE-KG-Triples | EDC+ Mistral-7b | 25.6 | 19.9 | 7.1 | 23.0 | 21.6 | 16.9 | 18.3 |
| | EDC+ Gemma-7b | 11.3 | 8.0 | 2.0 | 8.2 | 8.9 | 7.3 | 6.5 |
| | ReLiK RE | 3.2 | 4.6 | 2.7 | 3.8 | 4.1 | 4.0 | 4.4 |
| D-Triples | EDC+ Mistral-7b | 7.1 | 9.8 | 7.7 | 7.7 | 12.6 | 4.0 | 8.7 |
| | EDC+ Gemma-7b | 5.5 | 10.6 | 8.4 | 10.4 | 10.8 | 5.8 | 6.7 |

Table 3: *Recall* for the closed IE model ReLiK cIE (i.e., the extracted triples are linked to the KG) across KG snapshots, evaluated using the TKGU operations defined in Section 3.

| TKGU Operations | Model | 2019 | 2020 | 2021 | 2022 | 2023 | 2024 | 2025 |
|-----------------|-------|------|------|------|------|------|------|------|
| X-Triples | ReLiK cIE | 18.1 | 16.5 | 15.7 | 14.4 | 15.7 | 12.9 | 14.9 |
| E-Triples | ReLiK cIE | 14.9 | 16.8 | 14.0 | 13.4 | 15.2 | 12.5 | 14.7 |

## 5.2 METRICS AND EVALUATION

In order to evaluate the extraction and deprecation of triples based on emerging knowledge in text, we use recall as the primary metric (see Appendix B) to evaluate performance. We do not report precision or F1 scores in our main results Tables 2–3, as these metrics can be misleading under the open-world assumption (Razniewski et al., 2024). Under this assumption, the model may generate correct triple predictions that are incorrectly classified as false positives due to the inherently incomplete nature of KGs, which do not necessarily capture the full set of valid triples. We additionally provide precision and F1 scores in Tables 13–16 in Appendix F.1.

For models that do not link extracted triples to KG, as indicated in the column *KG Link* in Table 1 (ReLiK RE and EDC+), we evaluate recall with the *completeness score* (Jiang et al., 2024). This metric counts a ground-truth triple as correct if its cosine similarity with a predicted triple is above a set threshold (see Appendix B.2). This evaluation strategy is necessary because these models are not grounded in the entities present in the KG. This limitation underscores a broader research gap: existing IE methods operate largely independently of KG structure, making true text-driven KG updating challenging. Developing IE models that jointly exploit textual evidence and KG state to generate KG-grounded TKGU operations represents a promising direction for future work (see also Section K).

## 6 EXPERIMENTS AND ANALYSIS

Table 2 reports the performance of the ReLiK RE and EDC+ models across all TKGU operations. Table 3 shows the results for the ReLiK cIE model in the closed IE setting, which is restricted to TKGU operations involving existing entities and relations in the KG, namely *X-Triples* and *E-Triples*. The following paragraphs address key research questions and aim to lay the groundwork for future studies leveraging the TKGU operations introduced in this work.

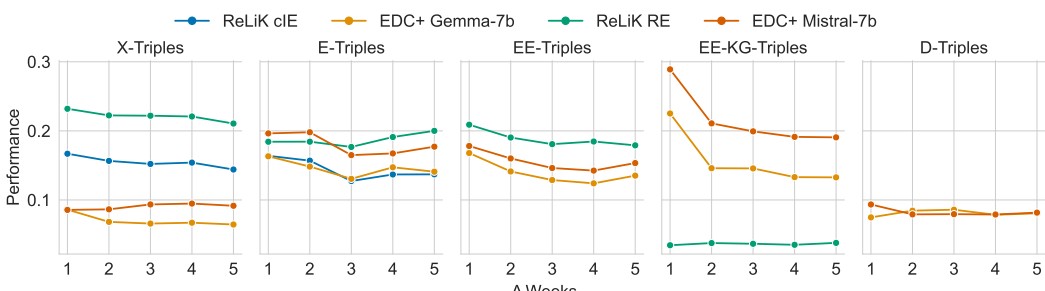

Figure 3: Performance of the models across temporal KG knowledge deltas. Some models show drops for certain TKGU operation types, for instance, EDC+Gemma-7b and EDC+Mistral-7b decline by over 5 percentage points between the first and second week deltas for EE-KG-Triples TKGU type.

**What is the general performance?** Overall, performance is low for both the recall metric reported in Table 3 and the completeness metric in Table 2. However, a closer inspection of the model predictions (see Appendix G) reveals that, in many cases, the extracted triples are semantically correct but do not align with the specific ground truth triples involved in the annotated TKGU operations. We hypothesize that this discrepancy arises because the models lack access to the KG content and structure, which prevents them from determining the nature of the knowledge being added and the types of relations involved. Access to KG-level statistics, such as the distribution of relation types, could provide valuable context and help improve model performance. This also points to a promising direction for future research: developing IE models that can identify emerging knowledge from unstructured text while leveraging the internal structure and temporal dynamics of the KGs.

**How do ReLiK and EDC+ differ in handling TKGU operations?** We selected the LLM-driven generative EDC+ model and the traditional, lightweight extractive span-based ReLiK model to compare how two fundamentally different and widely used architectures perform on TKGU operations. From Table 2, we observe that ReLiK RE significantly outperforms EDC+ on *X-Triples*. We hypothesize that this gap arises because ReLiK cIE and RE are explicitly trained to extract Wikidata triples from Wikipedia text, allowing the models to better capture relation structures and their distribution in the EMERGE corpora. In contrast, EDC+ relies only on a few in-context examples provided in the prompt, which appears insufficient to capture the diversity and complexity of relation types present in the dataset.

For TKGU operations that add previously non-existing triples to the KG, EDC+ performs comparably to ReLiK on *E-Triples* and *EE-Triples*. Furthermore, EDC+ significantly outperforms ReLiK RE on the *EE-KG-Triples* operation, which involves linking emerging entities mentioned in the passage to existing KG entities that are not explicitly referenced in the same passage. This result is expected, as ReLiK RE is designed to extract only entities explicitly mentioned in the text, as is also the case of other existing state-of-the-art IE models (see *EE-KG-Triples* column in Table 1). Its low performance on *EE-KG-Triples* is largely due to its reliance on explicit mentions: it extracts valid triples involving both emerging and existing entities that are present in the text but are not annotated in EMERGE, which includes only entity mentions explicitly annotated via Wikipedia hyperlinks.

When evaluated on *D-Triples*, EDC+ demonstrates relatively low performance, largely due to its lack of access to the knowledge graph. Without this information, the model cannot reliably identify triples that are already present and should be deprecated (see Appendix G for an example). In contrast, ReLiK is not trained to explicitly identify triples to be removed from KG and therefore is unable to extract D-Triples TKGU operation. This limitation also applies to other state-of-the-art IE models (see *D-Triples* column in Table 1).

**What is the performance across different snapshots?** Although results on earlier snapshots appear slightly higher than those from later years across different models and TKGU operations, there is no clear overall trend. We hypothesize that these performance differences are driven less by the

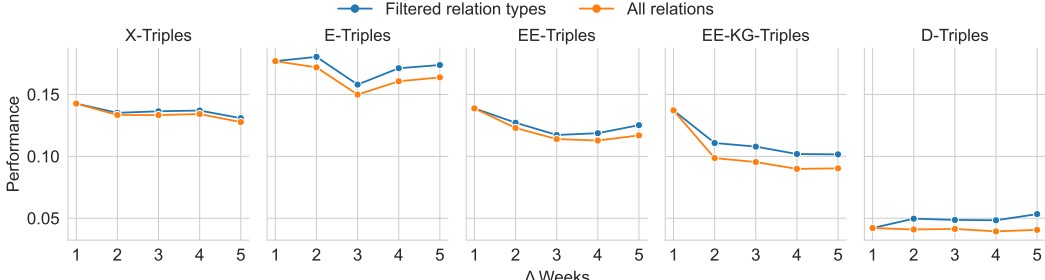

Figure 5: Performance of TKGU operations on relation types from the first KG delta week (*Filtered relations*) versus the full dataset including all relation types (*All relations*). The increased performance on *Filtered relations* shows that newly introduced relation types in later deltas are harder to predict, leading to larger performance drops.

novelty of the knowledge itself and more by the type of emerging knowledge dominant in each snapshot, an aspect we plan to investigate in future work.

**What is the performance on increasing temporal KG deltas?** In Figure 3, we plot model performance across increasing weekly KG deltas. Although not consistent across all TKGU operations and models, we generally observe a performance drop as deltas grow. We hypothesize that this decline stems from an increased number of relation types involved in the TKGU operations at higher deltas (see Figure 4). To test this hypothesis, we evaluate TKGU operations from the knowledge delta of week 2 onward while restricting relation types to those already present in week 1. Figure 5 shows the average performance difference across the evaluated models as the delta interval grows. Here, *Filtered relation types* denote performance restricted to relation types seen in week 1, while *All relations* corresponds to performance on the full set of relation types at each update. The reduced performance drop in the filtered setting supports our claim. In future work, we plan to further investigate this phenomenon and develop more robust models for continual knowledge updates under ever-increasing temporal deltas.

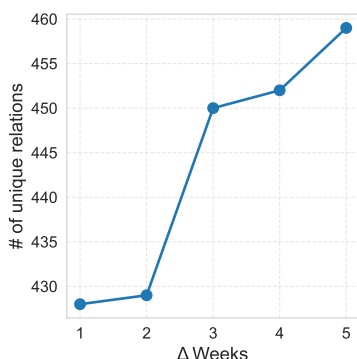

Figure 4: Evolution of the number of relation types with increasing weekly KG deltas.

## 7 CONCLUSION

In this work, we introduced EMERGE, the first dataset to cover all text-driven knowledge graph updating (TKGU) operations required to keep KGs aligned with emerging knowledge from textual sources. Evaluation of two state-of-the-art models on a dataset subset revealed a gap in current information extraction models to extract new information from text while accounting for existing KG content and structure. This suggests that future work should focus on designing methods capable of interacting with both emerging knowledge in text and the evolving content and structure of KGs. Additional limitations of our work, along with potential directions for future research, are discussed in Appendix K.

# 8 REPRODUCIBILITY STATEMENT

The code for dataset creation and reproducing the experimental results will be released in a public GitHub repository. The repository will also provide functionality for extending EMERGE with new KG snapshots, enabling incorporation of novel emerging knowledge (see Section 4.4). Moreover, the LLMs used for dataset annotation (Section 4.2) and within the EDC+ model (Section 5.1) are publicly accessible, enabling straightforward replication of dataset construction and experiments.

# 9 ETHICS STATEMENT

We confirm that we have read and adhered to the ICLR Code of Ethics throughout this work. Our study does not involve human subjects, personally identifiable information, or sensitive data (refer to Appendix L.2 for further details), and no ethical approval (e.g., IRB) was required. The datasets used are publicly available and comply with licensing and privacy requirements. We are not aware of any potential harms, security risks, or fairness concerns arising from the methods or applications of our research. There are no conflicts of interest, sponsorship influences, or legal compliance issues to disclose.

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

# A    EXTENDED RELATED WORK

This appendix provides an expanded discussion of related work, offering additional context, comparisons, and references beyond those included in the main text.

## A.1    COMPARISON OF EMERGE WITH EXISTING INFORMATION EXTRACTION BENCHMARKS

Table 4 presents a detailed comparison of EMERGE with existing information extraction (IE) benchmark datasets across the following key criteria:

- **Evolution:** indicates whether the dataset captures the natural evolution of knowledge in knowledge graph (*KG*) and textual (*Text*) sources.
- **Text-to-KG integration:** extent to which information extraction annotations are integrated with knowledge in a KG, broken down in:
  - **KG Link:** indicates whether the annotated entities in the triples are linked to a KG, supporting thus *entity linking* task.
  - **X-Triples:** presence of triples aligned with facts already in a KG (*X-Triples* TKGU operation; Section 3).
  - **E-Triples:** whether a dataset can be used to extract triples from text that connect existing entities in a KG (*E-Triples* TKGU type; Section 3).
  - **EE-Triples:** coverage of triples involving emerging (non-existing) entities in a KG (*EE-Triples* TKGU; Section 3).
  - **EE-KG-Triples:** availability of annotations linking emerging entities in text to other entities in KG not mentioned in text (*EE-KG-Triples* TKGU; Section 3).
  - **D-Triples:** inclusion of annotations that mark deprecation of existing KG triples based on information in textual passage (*D-Triples* TKGU; Section 3).

From Table 4, we observe that, to the best of our knowledge, none of the existing IE datasets support information extraction in a realistic knowledge evolution setting, where knowledge evolves simultaneously in both KG and textual sources (columns *Evolution-KG* and *Evolution-Text* in the table). Moreover, a number of datasets, such as TACRED (Zhang et al., 2017), BC5CDR (Li et al., 2016), DDI (Herrero-Zazo et al., 2013), and DWIE (Zaporojets et al., 2021), include *entity linking* to a KG, but are not accompanied by an actual KG, and their extracted relations do not align directly with the relations defined in a KG schema. Finally, although E-Triples and EE-Triples operations are nominally supported in some of the compared datasets, they do not capture genuinely emerging knowledge; instead, they rely on random subsampling of triples to approximate TKGU operations.

Table 4: Overview of major information extraction datasets from the past three decades across various domains, compared to our EMERGE dataset.

| | Evolution | | Text-to-KG integration | | | | | |
| Dataset | KG | Text | KG Link | X-Triples | E-Triples | EE-Triples | EE-KG-Triples | D-Triples |
|---|---|---|---|---|---|---|---|---|
| MUC-7 (1998) | ✗ | ✗ | ✗ | ✗ | ✗ | ✗ | ✗ | ✗ |
| CoNLL04 (2004) | ✗ | ✗ | ✗ | ✗ | ✗ | ✗ | ✗ | ✗ |
| ACE 2005 (2006) | ✗ | ✗ | ✗ | ✗ | ✗ | ✗ | ✗ | ✗ |
| SemEval 2010 (2010) | ✗ | ✗ | ✗ | ✗ | ✗ | ✗ | ✗ | ✗ |
| NYT (2010) | ✗ | ✗ | ✓ | ✓ | ✓ | ✗ | ✗ | ✗ |
| ADE (2012) | ✗ | ✗ | ✗ | ✗ | ✗ | ✗ | ✗ | ✗ |
| DDI (2013) | ✗ | ✗ | ✓ | ✗ | ✗ | ✗ | ✗ | ✗ |
| BC5CDR (2016) | ✗ | ✗ | ✓ | ✗ | ✗ | ✗ | ✗ | ✗ |
| WikiReading (2016) | ✗ | ✗ | ✗ | ✗ | ✗ | ✗ | ✗ | ✗ |
| ScienceIE(2017) | ✗ | ✗ | ✗ | ✗ | ✗ | ✗ | ✗ | ✗ |
| WebNLG (2017) | ✗ | ✗ | ✓ | ✓ | ✓ | ✗ | ✗ | ✗ |
| WNUT (2017) | ✗ | ✗ | ✗ | ✗ | ✗ | ✗ | ✗ | ✗ |
| TAC KBP (2017) | ✗ | ✗ | ✓ | ✓ | ✓ | ✓ | ✗ | ✗ |
| SciERC (2018) | ✗ | ✗ | ✗ | ✗ | ✗ | ✗ | ✗ | ✗ |
| TACRED (2017) | ✗ | ✗ | ✓ | ✗ | ✗ | ✗ | ✗ | ✗ |
| FewRel (2018) | ✗ | ✗ | ✓ | ✓ | ✓ | ✗ | ✗ | ✗ |
| FewRel 2.0 (2019) | ✗ | ✗ | ✓ | ✓ | ✓ | ✗ | ✗ | ✗ |
| Geo-NRE (2019) | ✗ | ✗ | ✓ | ✓ | ✓ | ✗ | ✗ | ✗ |
| Wiki-NRE (2019) | ✗ | ✗ | ✓ | ✓ | ✓ | ✗ | ✗ | ✗ |
| T-REX (2019) | ✗ | ✗ | ✓ | ✓ | ✓ | ✗ | ✗ | ✗ |
| DocRED (2019) | ✗ | ✗ | ✓ | ✓ | ✓ | ✗ | ✗ | ✗ |
| Wiki80 (2019) | ✗ | ✗ | ✓ | ✓ | ✓ | ✗ | ✗ | ✗ |
| FOBIE (2020) | ✗ | ✗ | ✗ | ✗ | ✗ | ✗ | ✗ | ✗ |
| DialogueRE (2020) | ✗ | ✗ | ✗ | ✗ | ✗ | ✗ | ✗ | ✗ |
| BioRel (2020) | ✗ | ✗ | ✓ | ✓ | ✓ | ✗ | ✗ | ✗ |
| Wiki20 (2020) | ✗ | ✗ | ✓ | ✓ | ✓ | ✗ | ✗ | ✗ |
| DWIE (2021) | ✗ | ✗ | ✓ | ✗ | ✗ | ✗ | ✗ | ✗ |
| KELM (2021) | ✗ | ✗ | ✓ | ✓ | ✓ | ✗ | ✗ | ✗ |
| REBEL (2021) | ✗ | ✗ | ✓ | ✓ | ✓ | ✗ | ✗ | ✗ |
| Re-TACRED (2021) | ✗ | ✗ | ✓ | ✗ | ✗ | ✗ | ✗ | ✗ |
| SMiLER (2021) | ✗ | ✗ | ✗ | ✗ | ✗ | ✗ | ✗ | ✗ |
| DrugProt (2021) | ✗ | ✗ | ✓ | ✗ | ✗ | ✗ | ✗ | ✗ |
| mLAMA (2021) | ✗ | ✗ | ✗ | ✗ | ✗ | ✗ | ✗ | ✗ |
| Re-DocRED (2022) | ✗ | ✗ | ✓ | ✓ | ✓ | ✗ | ✗ | ✗ |
| CDG (2022) | ✗ | ✗ | ✓ | ✓ | ✓ | ✗ | ✗ | ✗ |
| KD-DTI (2022) | ✗ | ✗ | ✓ | ✓ | ✓ | ✗ | ✗ | ✗ |
| FinRED (2022) | ✗ | ✗ | ✓ | ✓ | ✓ | ✗ | ✗ | ✗ |
| BioRED (2022) | ✗ | ✗ | ✓ | ✗ | ✗ | ✗ | ✗ | ✗ |
| SynthIE-text (2023) | ✗ | ✗ | ✓ | ✓ | ✓ | ✗ | ✗ | ✗ |
| REFinD (2023) | ✗ | ✗ | ✗ | ✗ | ✗ | ✗ | ✗ | ✗ |
| BioDEX (2023) | ✗ | ✗ | ✓ | ✗ | ✗ | ✗ | ✗ | ✗ |
| Text2KG (2023) | ✗ | ✗ | ✓ | ✓ | ✓ | ✗ | ✗ | ✗ |
| EMERGE (ours) | ✓ | ✓ | ✓ | ✓ | ✓ | ✓ | ✓ | ✓ |

## A.2 RELATED RESEARCH DIRECTIONS

We situate our contribution within four overlapping research directions outlined below, extending the related work described in Section 2.

**KG completion and refinement.** Research on KG completion (KGC) (Shen et al., 2022) and refinement (Paulheim, 2016; Subagdja et al., 2024) has led to the creation of a number of datasets

where the main task is to predict missing relations between entities. Thus, in their work, Bordes et al. (2013) introduced the WN18 and FB15k datasets. These datasets are derived from WordNet (Miller, 1995) and Freebase (Bollacker et al., 2008) respectively and capture the relations between entities. Later work (Toutanova & Chen, 2015; Dettmers et al., 2018) modified WN18 and FB15k datasets to eliminate redundant relations and train-test leakage, leading to the release of WN18RR and FB15K-237 datasets. More recently, a much larger Wikidata5M Wang et al. (2021) was released and contains $\sim 5$ million entities and $\sim 20$ million triples. Other widely used text-based KGC datasets are Wiki/NELL-One (Xiong et al., 2018), FB15K-237N (Lv et al., 2022), CoDEx (Safavi & Koutra, 2020), YAGO3-10 (Mahdisoltani et al., 2014) and LiterallyWikidata (Gesese et al., 2021). While these datasets enable models to incorporate textual information as node features (Daza et al., 2021; Wang et al., 2022), they remain static and do not capture the evolving nature of knowledge within KGs. Moreover, the KG triples in these datasets are not linked to textual sources that represent their information. To address this gap, our dataset captures the evolution of knowledge in the Wikidata KG and links KG updates to textual evidence from passages in Wikipedia pages. In doing so, it also creates opportunities to integrate ideas from KG completion, such as enforcing structural consistency, into text-driven information extraction (see next paragraph), thereby bridging two lines of work that are typically treated separately.

**Information extraction (IE).** To evaluate the ability of models to extract structured knowledge, researchers have developed IE datasets by annotating entity relations. MUC-7 (Chinchor & Marsh, 1998) introduced three relation types, with later datasets expanding in size, relation diversity, or both. Notable examples include CoNLL04 (Roth & Yih, 2004), ACE 2005 (Walker et al., 2006), ERE (Aguilar et al., 2014; Song et al., 2015), BC5CDR (Li et al., 2016), TACRED (Zhang et al., 2017), SciERC (Luan et al., 2018), SemEval-2010 (Hendrickx et al., 2010), SemEval-2017 (Augenstein et al., 2017), DWIE (Zaporojets et al., 2021) and BioRED (Luo et al., 2022), among others. Other datasets, such as NYT (Riedel et al., 2010), explicitly linked KG triples to textual snippets using distant supervision. Similarly, but on a larger scale, Gabrilovich et al. (2013) introduced FACC1 by aligning ClueWeb12 documents with Freebase entity mention annotations. In parallel, the TAC-KBP challenges (Ji et al., 2010; TAC-KBP, 2022) (2009 – 2020) produced proprietary manually annotated datasets for knowledge base population tasks such as slot filling and entity linking. More recently, these resources have been extended with a variety of datasets that map textual knowledge to KG literals, such as LiterallyWikidata (Gesese et al., 2021), and KG triples, such as WebNLG (Gardent et al., 2017), KELM (Agarwal et al., 2021), FewRel (Han et al., 2018), DocRED (Yao et al., 2019), Wiki/GEO-NRE (Distiawan et al., 2019), BioRel (Xing et al., 2020), T-REX (Elsahar et al., 2019) and REBEL (Cabot & Navigli, 2021). While these datasets connect textual knowledge to KG triples, they do not account for the operations required to update a KG as new information emerges in text. Our work addresses this gap by linking new textual knowledge to the specific update operations (see Section 3) on a KG snapshot. Methodologically, existing state-of-the-art IE methods (see Table 1) provide a natural starting point for tackling TKGU operations, as they extract structured knowledge from text and already cover some of these operations. However, they remain largely oblivious to the existing KG structure and require extensions to integrate emerging textual content into the KG, such as deprecating outdated triples, adding new entities, and enforcing structural consistency based on how entities and relations are used within the KG.

**Continual learning with emerging knowledge.** Over the last few years, there has been a growing interest in developing datasets aimed at probing models on emerging knowledge. Datasets like ECBD (Onoe et al., 2022), TemporalWiki (Jang et al., 2022a), TempLAMA (Dhingra et al., 2022), DynamicTempLAMA (Margatina et al., 2023), Updated and New LAMA (Jang et al., 2022b) were proposed to evaluate LLMs on slot-filling tasks using up-to-date knowledge. More recently, this line of work has expanded to question answering on emerging knowledge, with datasets such as StreamingQA (Liska et al., 2022), FreshQA (Vu et al., 2024), EvolvingQA (Kim et al., 2024b), RealtimeQA (Kasai et al., 2024), DynamicER (Kim et al., 2024a), GrowOVER (Ko et al., 2024), ERASE (Li et al., 2025a), Wiki-Update (Wu et al., 2024b), AntiLeak-Bench (Wu et al., 2024c), and LiveBench (White et al., 2024). However, existing datasets do not evaluate models on dynamically updating large-scale KGs while grounding changes in textual evidence. This setting requires models to be aware of changes in continually evolving KG schema and emerging knowledge in textual sources. To address this, we introduce EMERGE, a dataset that links emerging textual knowledge to updates in a time-evolving Wikidata KG with 37 million edges.

**KG versioning.** Our work is also related to KG versioning (Jin et al., 2025; Alam et al., 2024; Hofer et al., 2024). Similar to prior work in this area, we construct a KG in which each edge is annotated with its temporal span, capturing both its addition and deprecation history. This enables efficient extraction of KG snapshots and deltas (see Section 4.1). However, in contrast to KG versioning approaches that focus solely on maintaining temporal KG states, we use these versions as an intermediate step to build a dataset where KG updates are linked to the textual evidence in the corresponding Wikipedia passages. As a result, unlike existing IE datasets in which annotated triples are selected independently of KG evolution, the triples in EMERGE reflect the natural progression of facts in a real-world KG, making the dataset highly practical and grounded in authentic knowledge change.

# B    METRICS

## B.1    RECALL

We use recall, which measures the fraction of correctly predicted ground truth triples and is defined as follows:

$$\text{Recall} = \frac{|\mathcal{T}_{\mathcal{D}} \cap \mathcal{T}'_{\mathcal{D}}|}{|\mathcal{T}'_{\mathcal{D}}|},$$

where $\mathcal{T}_{\mathcal{D}}$ is a set of predicted triples and $\mathcal{T}'_{\mathcal{D}}$ is the set of ground truth triples.

## B.2    COMPLETENESS

The completeness metric (Jiang et al., 2024) can be formalized as follows:

$$c(\mathcal{T}'_{\mathcal{D}}, \mathcal{T}_{\mathcal{D}}) = \frac{|\{\tau \in \mathcal{T}'_{\mathcal{D}} | \exists \tau \in \mathcal{T}_{\mathcal{D}}, \text{sim}(\tau, \tau') \geq \phi\}|}{|\mathcal{T}'_{\mathcal{D}}|},$$

where $\mathcal{T}'_{\mathcal{D}}$ is the set of ground truth, and $\mathcal{T}_{\mathcal{D}}$ the set of predicted triples. $\text{sim}(\tau, \tau') = \text{CosSim}(emb(\tau), emb(\tau'))$. We use `SentenceTransformer('all-mpnet-base-v2')` to calculate the embeddings $emb$. We set the threshold $\phi$ to 0.9, which, based on our observations, provides accurate similarity matching.

# C    QUALITY CONTROL

In this section, we describe how LLMs are used to automatically filter out triples that cannot be derived from textual passages (Section C.1). We also detail the human annotation process used to validate the resulting LLM-generated annotations (Section C.2).

## C.1    QUALITY CONTROL PROMPTS AND EXAMPLES

We use two different prompts to filter out triples that cannot be inferred from a textual passage. The first is an *assertion prompt* (see Section C.1.1) applied to validate *X-Triples*, *E-Triples*, *EE-Triples*, and *EE-KG-Triples* as defined in Section 3. The goal of this prompt is to verify whether a triple can be directly or indirectly derived from the text. The second prompt is a *deprecation prompt* (see Section C.1.2), and is used to validate the deprecation of triples involved in *D-Triples* TKGU operation.

### C.1.1    TRIPLE ASSERTION PROMPT

The following is the structure of the prompt used to assert that the *X-Triples*, *E-Triples*, *EE-Triples*, and *EE-KG-Triples* TKGU operations can be derived from the information in textual passages. The placeholder `<TEXT>` is replaced by the textual passage, and `<TRIPLES_LIST>` by a list of triples.

```
You are given the following text:

<TEXT>
```

```
Can the following triples be directly or indirectly (the text
provides some hints) inferred from the text?  Use common sense
but not knowledge that cannot be inferred from the text above.

<TRIPLES_LIST>

Write a numbered list with the triples above, where each of the
triples is followed by YES if the triple is represented in the
text, and NO otherwise.  Follow your 'YES' or 'NO' answer for
each triple with a brief explanation.
```

The following is a concrete example of the *triple assertion* prompt outlined above:

```
You are given the following text:

After the war, Abraham H. Albertson returned to complete his
education and graduated, with scholarship assistance, from the
Columbia School of Architecture with a Ph.B in 1895.

Can the following triples be directly or indirectly (the text
provides some hints) inferred from the text?  Use common sense
but not knowledge that cannot be inferred from the text above.

1.  [Abraham H. Albertson(Q80520646); educated at(P69);
Columbia Graduate School of Architecture, Planning and
Preservation(Q5149701)]
2.  [Abraham H. Albertson(Q80520646); occupation(P106);
architect(Q42973)]
3.  [Abraham H. Albertson(Q80520646); country of
citizenship(P27); United States of America(Q30)]
4.  [Abraham H. Albertson(Q80520646); instance of(P31);
human(Q5)]
5.  [Abraham H. Albertson(Q80520646); archives at(P485);
University of Washington Libraries(Q7896575)]

Write a numbered list with the triples above, where each of
the triples is followed by YES if the triple is represented in
the text, and NO otherwise.  Follow your 'YES' or 'NO' answer
for each triple with a brief explanation.
```

Below is the response generated by `Meta-Llama-3.1-405B` when prompted with the example above:

```
1.  [Abraham H. Albertson(Q80520646); educated at(P69);
Columbia Graduate School of Architecture, Planning and
Preservation(Q5149701)] - YES - The text explicitly states
that Abraham H. Albertson graduated from the Columbia School
of Architecture.
2.  [Abraham H. Albertson(Q80520646); occupation(P106);
architect(Q42973)] - YES - Although the text does not directly
state that Abraham H. Albertson was an architect, it can be
inferred from the fact that he graduated from the Columbia
School of Architecture.
3.  [Abraham H. Albertson(Q80520646); country of
citizenship(P27); United States of America(Q30)] - NO -
The text does not provide any information about Abraham H.
Albertson's country of citizenship.
4.  [Abraham H. Albertson(Q80520646); instance of(P31);
```

```
human(Q5)] - YES - It can be inferred from the context that
Abraham H. Albertson is a human being, as he is described as
graduating from a school and having a war experience.
5. [Abraham H. Albertson(Q80520646); archives at(P485);
University of Washington Libraries(Q7896575)] - NO - The text
does not provide any information about the archives of Abraham
H. Albertson.
```

### C.1.2 PROMPT FOR TRIPLE DEPRECATION

The following prompt is used to verify whether the textual passage supports the deprecation of a set of triples from the KG:

```
You are given the following text:

<TEXT>

Does this text imply that the following triples are no longer
valid or do not hold at the moment the text was written?

<TRIPLES_LIST>

Write a numbered list with the triples above, where each of the
triples is followed by YES if the text implies that the triple
is no longer valid or do not hold, and NO otherwise. Follow
your 'YES' or 'NO' answer for each of the triples with a brief
explanation.
```

The following is a concrete example of the *triple deprecation* prompt outlined above:

```
You are given the following text:

Kate Lehrer (born Kate Tom Staples; December 17, 1937) is an
American writer, novelist and book reviewer from Washington,
D.C., and a panelist on the Diane Rehm Book Club on National
Public Radio. She was married to fellow writer and journalist
Jim Lehrer from 1960 until his death in 2020.

Does this text imply that the following triples are no longer
valid or do not hold at the moment the text was written?

1. [Kate Lehrer(Q6375599); spouse(P26); Jim Lehrer(Q931148)]
2. [Jim Lehrer(Q931148); spouse(P26); Kate Lehrer(Q6375599)]
3. [Jim Lehrer(Q931148); residence(P551); Washington,
D.C.(Q61)]

Write a numbered list with the triples above, where each of the
triples is followed by YES if the text implies that the triple
is no longer valid or do not hold, and NO otherwise. Follow
your 'YES' or 'NO' answer for each of the triples with a brief
explanation.
```

Below is the response generated by `Meta-Llama-3.1-405B` when prompted with the example above:

```
1. [Kate Lehrer(Q6375599); spouse(P26); Jim Lehrer(Q931148)]
- YES - The text states that Kate Lehrer was married to Jim
```

```
Lehrer ''until his death in 2020'', implying that the marriage
ended with his death, and therefore the spouse relationship is
no longer valid.
2.  [Jim Lehrer(Q931148); spouse(P26); Kate Lehrer(Q6375599)]
–  YES  – Similar to the previous triple, the text implies that
the marriage between Jim Lehrer and Kate Lehrer ended with his
death, so the spouse relationship is no longer valid.
3.  [Jim Lehrer(Q931148); residence(P551); Washington,
D.C.(Q61)]  –  NO  – The text does not provide any information
about Jim Lehrer's residence, let alone imply that he no longer
resides in Washington, D.C. It only mentions that Kate Lehrer
is from Washington, D.C.
```

## C.2 HUMAN ANNOTATION PROCESS

To evaluate the accuracy of automatic LLM annotations, we employ two expert annotators (researchers holding a PhD with a background in computer science) to manually label a subsample of 500 (100 triples per each TKGU type defined in Section 3) triples from 166 randomly selected passages from our EMERGE dataset. To ensure consistency and avoid ambiguity, the annotators follow a detailed set of guidelines described in Section C.2.1.

### C.2.1 ANNOTATION GUIDELINES

*Annotators were provided with the following guideline:*

For each annotation instance, you are provided with a textual passage, a triple, and an assessment type, which can be either *assert* or *deprecate*. For *assert* assessments, respond YES if the triple can be directly or indirectly inferred from the passage, and NO if it is not supported by the textual knowledge. For *deprecate* assessment, respond YES if the triple can be deprecated based on information present or implied in the passage, and NO otherwise. Take into account the following considerations when annotating for *assert* assessment type:

1. The triple may not be factually correct at the time the text was written, but it expresses a fact that holds true at some other point in time. For example, the triple ⟨*Barack Obama, president of, United States*⟩ should be assessed YES for the text passage "Barack Obama served as the 44th President of the United States from 2009 to 2017".

2. Use common world knowledge and reasoning to induce triples from textual passage. For example, the triple ⟨*Renault, headquarters in, France*⟩ should be assessed YES for the text passage "The headquarters of Renault are located in Boulogne-Billancourt, a suburb of Paris.", as Paris is located in France.

3. Mark with NO any concrete fact that cannot be inferred from text, even if some of the entities appear in the passage. For example, the triple ⟨*John Smith, participant in, Portland Climate Action Group protest*⟩ should be assessed NO for the passage "Several members of the Portland Climate Action Group gathered downtown to protest against deforestation and climate inaction.", as its factuality cannot be reliably inferred from the text.

4. Assess with NO the triples that cannot be reliably inferred from a textual passage. For example, the triple ⟨*David Bronkie, sibling, Eva Bronkie*⟩ should be assessed as NO for the passage: "David Bronkie and Eva Bronkie co-founded a sustainable home goods business focused on eco-friendly candle kits.", since the sibling relationship cannot be reliably inferred from the text (e.g., sharing the same last name).

Take into account the following considerations when annotating for *deprecate* assessment type:

1. The deprecation of a triple should be valid from the information provided in the passage and not the current status of the knowledge. For example, the triple ⟨ *Donald Trump, president of, United States* ⟩ should be assessed with YES for the passage "Joe

Biden is the President of the United States, having taken office recently and begun his tenure with notable public appearances and speeches.", despite the fact that Donald Trump may be a current president of United States.

2. The deprecation of a triple might not be explicitly stated in the text, but can be implied. For example, the deprecation of the triple ⟨*Hans Rausing, spouse, Julia Rausing*⟩ should be assessed as YES for the passage "Julia Rausing, the philanthropist and business heiress, passed away on April 18, 2024, at the age of 63 after a long battle with cancer. She is survived by her husband, Hans Rausing, and their family.", since the marital relationship is no longer current due to Julia Rausing's death, which implies that the triple is deprecated.

3. Assess with NO any triples whose deprecation can not be reliably inferred from text, even if some of the entities appear in the text.

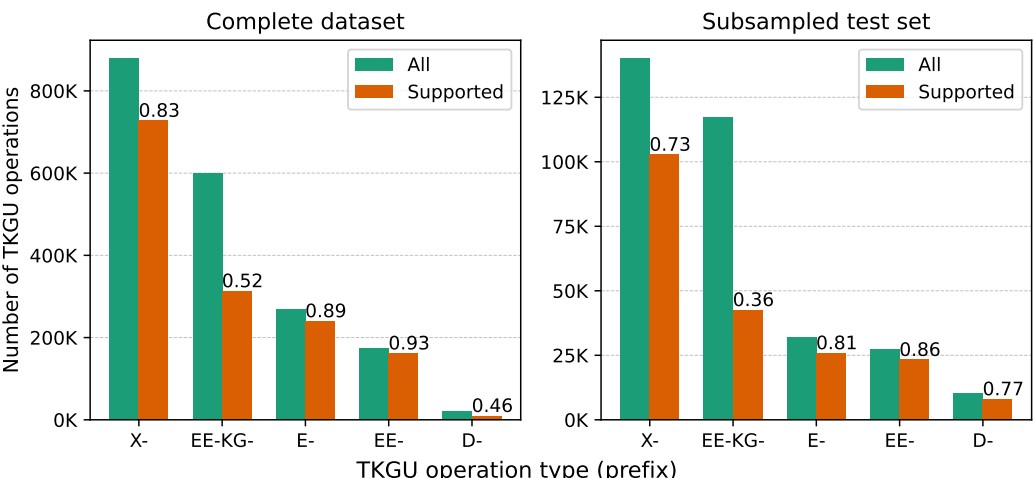

Figure 6: The ratio of TKGU operations supported by the LLM to the total number of TKGU operations mapped to textual passages during the alignment process.

### C.2.2 ANNOTATION AGREEMENT

We report annotation agreement between the two human annotators (*H–H Cohen's $\kappa$*), as well as between each human annotator and the LLM (*H1–LLM Cohen's $\kappa$* and *H2–LLM Cohen's $\kappa$*) in Table 5. The Cohen's $\kappa$ scores indicate strong agreement (0.6–0.8) to almost perfect agreement ($>$ 0.8). In addition, we compute Fleiss' $\kappa$ (*H+LLM Fleiss' $\kappa$*) and Krippendorff's $\alpha$ (*H+LLM Kripp. $\alpha$*) to assess agreement among all three annotators, both humans and the LLM. Consistent with Cohen's $\kappa$, these metrics also show strong to almost perfect agreement. This supports the use of the evaluated `Meta-Llama-3.1-405B` LLM to annotate full dataset using the prompts described in the Appendix C.1.

### C.3 TRIPLE ANNOTATION STATISTICS

Figure 6 illustrates the ratio of triples aligned with textual passages during the *alignment* step described in Section 4.2 that were marked by automatic LLM annotations – using the prompts detailed in Section C.1 – as not representative of the passages. This ratio is different between the *complete* and *subsampled* dataset used during testing. The reason is that during subsampling we retain instances with supported by LLM D-Triples operations (see Section 5). Additionally, we observe a lower fraction of EE-KG-Triples supported by the LLM. This occurs because EE-KG-Triples include all entities in the KG, many of which are unrelated to the passage content but are connected to emerging entities mentioned in the text. Consequently, these triples are inherently less likely

Table 5: Annotation agreement per TKGU operation and overall. Columns show pairwise Cohen's $\kappa$ between humans (H-H) and between each human and the LLM (H1-LLM, H2-LLM), as well as multi-rater agreement including all three annotators (H+LLM) measured with Fleiss' $\kappa$ and Krippendorff's $\alpha$.

| TKGU Operation | H-H Cohen's $\kappa$ | H1-LLM Cohen's $\kappa$ | H2-LLM Cohen's $\kappa$ | H+LLM Fleiss' $\kappa$ | H+LLM Kripp. $\alpha$ |
|---|---|---|---|---|---|
| X-Triples | 0.718 | 0.649 | 0.637 | 0.668 | 0.669 |
| E-Triples | 0.750 | 0.698 | 0.750 | 0.732 | 0.733 |
| EE-Triples | 0.680 | 0.811 | 0.863 | 0.784 | 0.785 |
| EE-KG-Triples | 0.880 | 0.840 | 0.761 | 0.827 | 0.827 |
| D-Triples | 0.771 | 0.675 | 0.610 | 0.687 | 0.688 |
| Overall | 0.792 | 0.765 | 0.744 | 0.767 | 0.767 |

Table 6: Statistics of our newly introduced EMERGE dataset, organized by KG snapshots (rows). For each snapshot, we report the number of *instances* and TKGU *operations* in both the *complete dataset* and the *subsampled test set*. The *KG statistics* section summarizes the number of entities, relation types, and triples in each KG snapshot.

| Snapshot | Complete dataset | | Subsampled test set | | KG statistics | | |
|---|---|---|---|---|---|---|---|
| | Instances | Operations | Instances | Operations | Entities | Rel. Types | Triples |
| 2019 | 37K | 202K | 5K | 24K | 5.96M | 5,646 | 25.73M |
| 2020 | 31K | 199K | 5K | 26K | 6.14M | 7,017 | 28.76M |
| 2021 | 40K | 292K | 5K | 36K | 6.34M | 8,216 | 30.84M |
| 2022 | 30K | 188K | 5K | 27K | 6.54M | 9,425 | 33.41M |
| 2023 | 26K | 151K | 5K | 26K | 6.67M | 10,599 | 34.99M |
| 2024 | 32K | 200K | 5K | 29K | 6.80M | 11,409 | 36.31M |
| 2025 | 33K | 217K | 5K | 31K | 6.93M | 12,304 | 37.54M |

to be supported by the passages. A promising future direction is to develop information extraction methods that rely not only on textual evidence to extract triples but also integrate this content with existing knowledge and patterns in the KG. Such an approach could be particularly beneficial for incorporating emerging entities in EE-KG-Triples, even when they are not supported by textual passages, into the broader KG.

## D  DATASET STATISTICS

In this section we will present additional statistics of EMERGE.

### D.1  OVERALL STATISTICS OF EMERGE

Table 6 presents key statistics of our newly introduced EMERGE dataset, broken down by KG reference snapshots. For each snapshot, we report the number of *instances* and TKGU *operations* in both the full dataset and the subsampled test set. The table also summarizes *KG* snapshots statistics, including the number of entities, relation types, and triples in each snapshot. We observe that the number of entities, relation types, and triples increases over time, reflecting the growth of Wikidata and the addition of new relations to the KG schema. This evolving structure creates a challenging scenario for future models, which must recognize these changes in the KG and adapt their predictions accordingly.

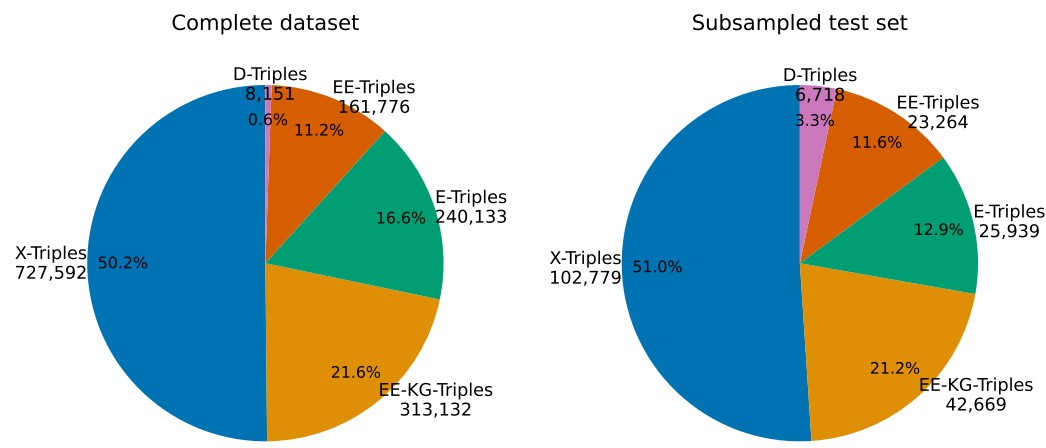

Figure 7: Distribution of TKGU operations defined in Section 3 in EMERGE. The left subgraph shows the full dataset, while the right one shows the subsampled test set (see Section 5). In the test set, D-Triples are retained at higher frequency to ensure sufficient evaluation, while other TKGU operation types reflect the original dataset distribution.

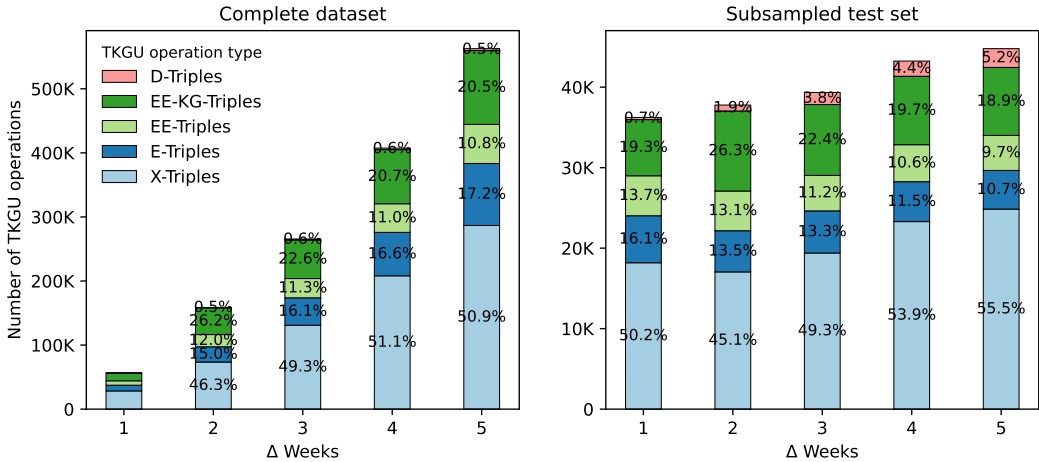

Figure 8: Distribution of TKGU operations across KG deltas up to 5 weeks defined in EMERGE.

## D.2 NUMBER OF TKGU OPERATIONS AND THEIR DISTRIBUTION

Figure 7 illustrates the distribution of KG update operations for each TKGU type defined in Section 3. We report on both the complete dataset (left subgraph) and the subsampled test set (right subgraph). Furthermore, we display both the number as well as the percentage the operations of each of the TKGU types represent in EMERGE. This distribution is very similar between the complete dataset and subsampled test set, except for D-Triples, which were retained at higher frequency in the test set to ensure sufficient evaluation (see Section 5). Additionally, Figure 8 shows the distribution of TKGU update operations across temporally increasing weekly KG deltas. In the *Complete dataset* (left subplot), the number of TKGU operations increases with larger deltas. A similar trend is visible in the *subsampled test set* (right subplot), although the growth is less pronounced. This is due to our subsampling procedure, which retains only 1,000 instances per delta (see Section 5), resulting in a more uniform distribution of operations across deltas.

# E QUALITATIVE ANALYSIS

In this section, in Tables 7–12 we present the five frequent factual triples from EMERGE for each of the TKGU operation types, with an example of corresponding textual passage. The goal is to highlight representative cases that illustrate both the contents of the benchmark and the challenges it poses. The information in the tables contains the KG snapshot (*Snap.*) used to compute weekly knowledge deltas aligned with each passage. We also report the number of occurrences of the triple in the *Triple* column within EMERGE (#), along with an example passage. The emerging entities in TKGU operations appear in bold. Due to space constraints, we selected the shortest passages; however, in EMERGE, passages consist of full Wikipedia paragraphs.

Our main observation is that the derived TKGU operations are closely aligned with the primary events occurring immediately after each KG snapshot (all snapshots are taken on January 1st of the corresponding year). We also note that the resulting triples are highly specific to the Wikidata KG structure. This is particularly evident in Table 10, which shows examples of EE-KG-Triples, where an emerging entity must be connected to the existing KG. Consequently, we believe a promising future direction is to develop information extraction models that consider KG structure when proposing knowledge updates in it.

Additionally, to illustrate the effectiveness of using an LLM (`Meta-Llama-3.1-405B`) to verify that all TKGU operations can be derived from their corresponding textual passages during the *curation* step described in Section 4.2, we present the most frequent factual triples from the EE-KG-Triples TKGU operation in EMERGE that were marked as *not supported* by the LLM in Table 11. These examples highlight triples that occurred frequently but were flagged because the LLM determined that their source textual passages did not support them. None of these triples are grounded in the corresponding text, demonstrating the reliability of the LLM-based validation process.

Table 7: Example entries of the most frequent X-Triples TKGU operation instances in EMERGE, showing the snapshot (Snap.), triple, and number of instances (#).

| Snap. | # | Triple | Example Passage |
|---|---|---|---|
| 2021 | 834 | ⟨Donald Trump; candidacy in election; 2020 United States presidential election⟩ | Over the span of the 2020 presidential election, RSBN's coverage of Donald Trump's campaign rallies grossed over 127 million views on YouTube. |
| 2021 | 827 | ⟨2020 United States presidential election; candidate; Donald Trump⟩ | In 2020, Pletts voiced support for Donald Trump and the Republican Party in the 2020 United States presidential election and Senate elections. |
| 2021 | 671 | ⟨Joe Biden; candidacy in election; 2020 United States presidential election⟩ | In September 2020, Kennedy Kent endorsed Republican President Donald Trump for re-election over Democratic nominee Joe Biden. |
| 2021 | 666 | ⟨2020 United States presidential election; candidate; Joe Biden⟩ | Despite being divorced, she remains good friends with her ex-husband, and she supported Joe Biden and Kamala Harris in the 2020 election. |
| 2021 | 586 | ⟨midfielder; sport; association football⟩ | "Niko Rak" (born 26 July 2003) is a Croatian footballer who plays for Šibenik as a midfielder. |

Table 8: Example entries of the most frequent E-Triples TKGU operation instances in EMERGE, showing the snapshot (Snap.), triple, and number of instances (#).

| Snap. | # | Triple | Example Passage |
|---|---|---|---|
| 2021 | 315 | ⟨Joe Biden; position held; President of the United States⟩ | On 20 January 2021, Joe Biden was sworn in as 46th President of the United States. |
| 2023 | 204 | ⟨Kevin McCarthy; position held; Speaker of the United States House of Representatives⟩ | On January 3, 2023, at the beginning of the 118th Congress, Boebert voted for Jim Jordan to be the U.S. House Speaker, in rebuke of House Minority Leader Kevin McCarthy. |
| 2020 | 168 | ⟨Abu Mahdi al-Muhandis; military branch; Popular Mobilization Forces⟩ | Abu Mahdi al-Muhandis returned to Iraq following the withdrawal of US troops (December 2011) to head the Kata'ib Hezbollah militia,; he then became deputy chief of the Popular Mobilization Forces. |
| 2024 | 164 | ⟨Houthi movement; country; Yemen⟩ | On 28 March 2021, the Houthis forced 13 Jews to leave Yemen, they only allowed four elderly Jews to live in Yemen. |
| 2020 | 138 | ⟨Qasem Soleimani; place of death; Baghdad⟩ | Soleimani was assassinated in a targeted U.S. drone strike on 3 January 2020 in Baghdad, which was approved by President Donald Trump on the grounds that Soleimani posed an "imminent threat" to American lives. |

Table 9: Example entries of the most frequent EE-Triples TKGU operation instances in EMERGE (emerging entities in bold), showing the snapshot (Snap.), triple, and number of instances (#).

| Snap. | # | Triple | Example Passage |
|---|---|---|---|
| 2021 | 848 | ⟨**January 6 United States Capitol attack**; significant person; Donald Trump⟩ | She called for the impeachment of President Donald Trump, in wake of the 2021 storming of the United States Capitol. |
| 2020 | 670 | ⟨Qasem Soleimani; significant event; **assassination of Qasem Soleimani**⟩ | He was killed by a targeted U.S. drone strike at the Baghdad International Airport on 3 January 2020, which also killed Iranian Armed Forces Major General Qasem Soleimani. |
| 2022 | 317 | ⟨**Dawn FM**; performer; The Weeknd⟩ | In 2022 the group also received credit for co producing songs off The Weeknds fifth studio album Dawn FM. |
| 2025 | 291 | ⟨**2025 New Orleans truck attack**; located in the administrative territorial entity; New Orleans⟩ | 2025 New Orleans truck attack: President Joe Biden has been briefed on the attack and has been in touch with New Orleans Mayor to offer support. |
| 2023 | 72 | ⟨**Flowers**; performer; Miley Cyrus⟩ | The chart's current number one as of the issue dated January 28, 2023, is "Flowers" by Miley Cyrus |

Table 10: Example entries of the most frequent EE-KG-Triples TKGU operation instances in EMERGE (emerging entities in bold), showing the snapshot (Snap.), triple, and number of instances (#).

| Snap. | # | Triple | Example Passage |
|---|---|---|---|
| 2021 | 3149 | ⟨**January 6 United States Capitol attack**; located in the administrative territorial entity; Washington, D.C.⟩ | January 6 United States Capitol attack: The Proud Boys posted messages boasting and taking credit for causing "absolute terror". |
| 2020 | 1097 | ⟨**assassination of Qasem Soleimani**; instance of; assassination⟩ | Assassination of Qasem Soleimani: the president called for restraint and said the events in Iraq were the result of previous "terrorist acts". |
| 2025 | 991 | ⟨**Golden Age of Argentine cinema**; part of; history of film⟩ | "Volver a vivir" is a 1941 Argentine film of the Golden Age of Argentine cinema. |
| 2024 | 282 | ⟨**South Africa v. Israel**; charge; genocide⟩ | In 2023-24, he was appointed as a member of the South African legal team arguing "South Africa v. Israel" regarding the Genocide Convention. |
| 2019 | 179 | ⟨**All Elite Wrestling**; instance of; business⟩ | On January 1, 2019 Cody Rhodes unveiled a new promotion; All Elite Wrestling, in which he, along with Matt and Nick Jackson, will serve as Executive Vice President. |

Table 11: Representative examples of the most frequent EE-KG-Triples TKGU operation instances in EMERGE **filtered out by the `Meta-Llama-3.1-405B` curator**. Emerging entities are in bold. Each row shows the snapshot (Snap.), triple, and number of occurrences (#). None of these triples are supported by the corresponding textual passages, illustrating the effectiveness of the LLM-based filtering.

| Snap. | # | Triple | Example Passage |
|-------|-----|--------|-----------------|
| 2021 | 1174 | ⟨Proud Boys; significant event; **January 6 United States Capitol attack**⟩ | Trump supporters infiltrated Capitol Hill in Washington DC., 5 people killed. |
| 2022 | 158 | ⟨**Dawn FM**; distribution format; LP record⟩ | In 2022 the group also received credit for co producing songs off The Weeknds fifth studio album Dawn FM. |
| 2024 | 123 | ⟨**2024 Haneda Airport runway collision**; destination point; Niigata Airport⟩ | 2024 Haneda Airport runway collision: All flights in and out of Haneda were suspended following the accident; operations currently remain suspended. |
| 2019 | 55 | ⟨**All Elite Wrestling**; legal form; privately held company⟩ | On January 1, 2019 Cody Rhodes unveiled a new promotion; All Elite Wrestling, in which he, along with Matt and Nick Jackson, will serve as Executive Vice President. |
| 2019 | 29 | ⟨**@world_record_egg**; country; United Kingdom⟩ | @world_record_egg is an account on social media platform Instagram, notable for holding the world records for both the most-liked Instagram post and most liked online post on any media platform in history. |

Table 12: Example entries of the most frequent D-Triples TKGU operation instances in EMERGE (emerging entities in bold), showing the snapshot (Snap.), triple, and number of instances (#).

| Snap. | # | Triple | Example Passage |
|---|---|---|---|
| 2024 | 88 | ⟨**Adam Peters**; member of sports team; San Francisco 49ers⟩ | Peters joined the Denver Broncos as a scout in 2009. He was promoted to assistant director of college scouting in July 2014 and to director of college scouting in 2016. He was a member of the team that won Super Bowl 50 in 2015. |
| 2021 | 79 | ⟨Parler; distributed by; Google Play⟩ | After complaints that Parler was used to coordinate the 2021 storming of the U.S. Capitol, Apple and Google removed Parler's mobile app from their app stores. Parler went offline on January 10, 2021 at 11:59 PM (PST) after Amazon Web Services canceled its hosting services. |
| 2021 | 75 | ⟨Mike Pence; position; Vice President of the United States⟩ | "Marlon Bundo", also known as "Bunny of the United States" ("BOTUS"), is a rabbit, belonging to the family of Mike Pence, the 48th and former Vice President of the United States. |
| 2025 | 63 | ⟨Vice President of the United States; position holder; Kamala Harris⟩ | West is the brother-in-law of former Vice President Kamala Harris. He served as an advisor to her 2024 presidential campaign. |
| 2020 | 43 | ⟨European Union; has part(s); United Kingdom⟩ | Chris Davies was the chairman (2019 – 2020) - until the United Kingdom left the European Union. |

# F ADDITIONAL EXPERIMENTAL RESULTS

## F.1 PRECISION AND F1 SCORES

In this section, we present precision and F1 results for both *closed information extraction* (ReLiK cIE; Tables 13–14) and *relation extraction* (ReLiK RE and EDC+; Tables 15–16). Closed information extraction performance is measured via exact triple matching against the ground truth. Relation extraction is evaluated using the completeness-score approximation with a threshold $\phi = 0.9$ (see Section B.2 for the formal definition).

Table 13: *Precision* for the closed IE model ReLiK cIE (i.e., the extracted triples are linked to the KG) across KG snapshots, evaluated using the TKGU operations defined in Section 3.

| TKGU | Model | 2019 | 2020 | 2021 | 2022 | 2023 | 2024 | 2025 |
|---|---|---|---|---|---|---|---|---|
| X-Triples | ReLiK cIE | 45.8 | 31.4 | 35.1 | 37.2 | 33.3 | 34.0 | 31.2 |
| E-Triples | ReLiK cIE | 2.6 | 3.2 | 1.9 | 2.9 | 3.3 | 2.7 | 2.7 |

Table 14: *F1 score* for the closed IE model ReLiK cIE (i.e., the extracted triples are linked to the KG) across KG snapshots, evaluated using the TKGU operations defined in Section 3.

| TKGU | Model | 2019 | 2020 | 2021 | 2022 | 2023 | 2024 | 2025 |
|---|---|---|---|---|---|---|---|---|
| X-Triples | ReLiK cIE | 25.9 | 21.5 | 21.6 | 20.5 | 21.2 | 18.6 | 20.1 |
| E-Triples | ReLiK cIE | 4.5 | 5.4 | 3.3 | 4.8 | 5.5 | 4.4 | 4.6 |

Table 15: *Precision* (measured using the completeness score) for IE models that do not link extracted triples to the KG, evaluated across KG snapshots on the TKGU operations defined in Section 3.

| TKGU | Model | 2019 | 2020 | 2021 | 2022 | 2023 | 2024 | 2025 |
|---|---|---|---|---|---|---|---|---|
| | EDC+ Mistral-7b | 4.7 | 3.2 | 4.9 | 4.1 | 5.2 | 3.7 | 4.9 |
| X-Triples | EDC+ Gemma-7b | 3.2 | 2.8 | 3.1 | 2.5 | 3.2 | 2.5 | 3.4 |
| | ReLiK RE | 9.2 | 8.2 | 8.1 | 9.2 | 8.6 | 9.2 | 9.9 |
| | EDC+ Mistral-7b | 2.5 | 2.3 | 1.9 | 2.3 | 2.4 | 2.3 | 2.1 |
| E-Triples | EDC+ Gemma-7b | 2.0 | 1.5 | 1.3 | 1.6 | 1.8 | 1.5 | 1.5 |
| | ReLiK RE | 2.4 | 2.0 | 1.9 | 1.9 | 1.9 | 1.7 | 1.5 |
| | EDC+ Mistral-7b | 1.9 | 2.3 | 1.1 | 1.8 | 2.1 | 1.6 | 1.8 |
| EE-Triples | EDC+ Gemma-7b | 1.4 | 1.5 | 0.9 | 1.4 | 1.7 | 1.3 | 1.4 |
| | ReLiK RE | 1.7 | 1.9 | 1.0 | 2.4 | 2.2 | 1.7 | 1.6 |
| | EDC+ Mistral-7b | 1.6 | 2.1 | 1.8 | 1.8 | 1.9 | 1.5 | 1.8 |
| EE-KG-Triples | EDC+ Gemma-7b | 2.2 | 2.7 | 1.6 | 1.9 | 2.4 | 2.0 | 2.0 |
| | ReLiK RE | 0.3 | 0.7 | 0.8 | 0.5 | 0.6 | 0.6 | 0.7 |
| D-Triples | EDC+ Mistral-7b | 8.5 | 9.8 | 14.4 | 9.6 | 15.5 | 5.8 | 11.6 |
| | EDC+ Gemma-7b | 5.0 | 6.9 | 8.3 | 8.9 | 8.6 | 3.3 | 6.4 |

Table 16: *F1 score* (measured using the completeness score) for IE models that do not link extracted triples to the KG, evaluated across KG snapshots on the TKGU operations defined in Section 3.

| TKGU | Model | 2019 | 2020 | 2021 | 2022 | 2023 | 2024 | 2025 |
|---|---|---|---|---|---|---|---|---|
| | EDC+ Mistral-7b | 6.3 | 4.5 | 6.7 | 5.4 | 7.2 | 5.0 | 6.3 |
| X-Triples | EDC+ Gemma-7b | 4.5 | 4.2 | 4.4 | 3.5 | 4.5 | 3.5 | 4.5 |
| | ReLiK RE | 13.5 | 12.3 | 12.1 | 12.6 | 12.4 | 12.4 | 13.3 |
| | EDC+ Mistral-7b | 4.5 | 4.0 | 3.3 | 4.0 | 4.3 | 4.2 | 3.8 |
| E-Triples | EDC+ Gemma-7b | 3.5 | 2.8 | 2.3 | 2.8 | 3.2 | 2.8 | 2.7 |
| | ReLiK RE | 4.3 | 3.7 | 3.4 | 3.4 | 3.4 | 3.0 | 2.8 |
| | EDC+ Mistral-7b | 3.4 | 4.0 | 2.0 | 3.2 | 3.7 | 2.8 | 3.2 |
| EE-Triples | EDC+ Gemma-7b | 2.6 | 2.7 | 1.6 | 2.6 | 3.1 | 2.4 | 2.6 |
| | ReLiK RE | 3.1 | 3.5 | 1.8 | 4.4 | 4.0 | 3.1 | 2.9 |
| | EDC+ Mistral-7b | 3.0 | 3.7 | 2.9 | 3.3 | 3.5 | 2.8 | 3.4 |
| EE-KG-Triples | EDC+ Gemma-7b | 3.7 | 4.0 | 1.8 | 3.1 | 3.8 | 3.1 | 3.0 |
| | ReLiK RE | 0.5 | 1.3 | 1.3 | 0.8 | 1.0 | 1.1 | 1.2 |
| D-Triples | EDC+ Mistral-7b | 7.8 | 9.8 | 10.0 | 8.5 | 13.9 | 4.7 | 9.9 |
| | EDC+ Gemma-7b | 5.2 | 8.4 | 8.3 | 9.6 | 9.6 | 4.2 | 6.6 |

## G  QUALITATIVE ANALYSIS OF THE RESULTS

Table 17 presents three instances from EMERGE, together with the predictions of the evaluated models described in Section 5.1. We observe that while most model predictions are semantically correct, they often do not correspond to the TKGU operations that capture actual changes in the KG. For instance, in *passage 1*, the predicted *D-Triples* are reasonable but differ from the triple actually deprecated in the KG: ⟨CLC; has part; Elkie⟩. This discrepancy is expected, as current state-of-the-art information extraction models are largely unaware of the structure and content of KGs (see Table 1). We observe similar mismatches for the X/E/EE/EE-KG-Triples operations in *passage 2* and *passage 3*. Moreover, the LLM-driven EDC+ method exhibits a high degree of hallucination, particularly for **EE-KG-Triples**, where an emerging entity must be connected to the KG. Since EDC+ does not interact with the KG, its predicted emerging entities often fail to correspond to the actual ones. For instance, in *passage 3*, the true emerging entity is Puttsburg Maulers, yet both *EDC+ Mistral-7b* and *EDC+ Gemma-7b* instead generate EE-KG-Triples around Kirby Wilson. In addition, many of the generated triples are not supported by the passage itself. We hypothesize

that enabling these models to interact with KG content and structure could mitigate such errors, as predictions would then be grounded not only in the text but also aligned with existing KG contents.

Table 17: Table showcasing three instances (passages aligned to Ground truth TKGU operations) of EMERGE. We also display the predictions of the models used in our experiments (see Section 5.1). While these predictions are often correct, they frequently fail to align with the existing knowledge and structure of the KG.

| | |
|---|---|
| **Passage 1** | On February 3, 2021, Cube Entertainment confirmed Elkie's departure from CLC, and her contract with the company has been terminated. |
| **Ground truth** | **D-Triples**: ⟨CLC; has part; Elkie⟩ |
| **EDC+ Mistral-7b** | **D-Triples**: ⟨Elkie; member of musical group; CLC⟩ |
| **EDC+ Gemma-7b** | **D-Triples**: ⟨CLC; employee; Elkie⟩ |
| **ReLiK RE** | – |
| **ReLiK cIE** | – |
| **Passage 2** | "Cancilla liliformis" is a species of sea snail, a marine gastropod mollusk, in the family Mitridae, the miters or miter snails. |
| **Ground truth** | **EE-Triples**: ⟨Cancilla liliformis; taxon rank; species⟩
**X-Triples**: ⟨Mitridae; taxon rank; family⟩, ⟨Gastropoda; parent taxon; Mollusca⟩
**EE-KG-Triples**: ⟨Cancilla liliformis; instance of; taxon⟩ |
| **EDC+ Mistral-7b** | **X/E/EE-Triples**: ⟨Cancilla liliformis; species of; Mitridae⟩
**EE-KG-Triples**: ⟨Cancilla liliformis; instance of; marine organism⟩, ⟨Cancilla liliformis; instance of; mollusk⟩ |
| **EDC+ Gemma-7b** | **X/E/EE-Triples**: ⟨Cancilla liliformis; species of; Mitridae⟩ ⟨Cancilla liliformis, instance of, Gastropoda⟩
**EE-KG-Triples**: ⟨Cancilla liliformis, country of citizenship, France⟩, ⟨Cancilla liliformis, place of birth, France⟩, ⟨Cancilla liliformis, instance of, Human⟩ |
| **ReLiK RE** | **X/E/EE/EE-KG-Triples**: ⟨Cancillopsis liliformis; subclass of; Mitridae⟩, ⟨Cancilla liliformis; subclass of; Mitridae⟩ |
| **ReLiK cIE** | **X/E/EE/EE-KG-Triples**: ⟨Sea snail; subclass of; gastropod⟩, ⟨Mitridae; subclass of; gastropod⟩ |
| **Passage 3** | Pittsburgh Maulers (USFL): On January 20, 2022, Wilson was named Head Coach and General Manager of the Pittsburgh Maulers of the United States Football League (USFL). |
| **Ground truth** | **EE-Triples**: ⟨Pittsburgh Maulers; league; United States Football League⟩
**EE-KG-Triples**: ⟨Pittsburgh Maulers; country; United States of America⟩, ⟨Pittsburgh Maulers; sport; American football⟩ |
| **EDC+ Mistral-7b** | **X/E/EE-Triples**: ⟨Pittsburgh Maulers, head coach, Kirby Wilson⟩
**EE-KG-Triples**: ⟨Kirby Wilson; educated at; University of Pittsburgh⟩, ⟨Kirby Wilson; place of birth, Pittsburgh⟩, ⟨Kirby Wilson; country of citizenship; United States⟩, ⟨Kirby Wilson; instance of; human⟩ |
| **EDC+ Gemma-7b** | **X/E/EE-Triples**: ⟨Pittsburgh Maulers; head coach; Kirby Wilson⟩, ⟨Pittsburgh Maulers; country; United States⟩
**EE-KG-Triples**: ⟨Kirby Wilson; educated at; University of North Texas⟩, ⟨Kirby Wilson; place of birth; Dallas⟩, ⟨Kirby Wilson; instance of; human⟩ |
| **ReLiK RE** | **X/E/EE/EE-KG-Triples**: ⟨Kirby Wilson; member of sports team; Pittsburgh Maulers⟩, ⟨Kirby Wilson; member of; Pittsburgh Maulers⟩, ⟨Kirby Wilson; member of sports team; Pittsburgh Maulers⟩ |
| **ReLiK cIE** | ⟨Kirby Wilson; member of sports team; Pittsburgh Maulers⟩, ⟨Wilson; member of sports team; Pittsburgh Maulers⟩, ⟨Head Coach; member of sports team; Pittsburgh Maulers⟩, ⟨Pittsburgh Maulers; organizer; USFL⟩ |

## H EDC+ EXECUTION

### H.1 EDC+ EXECUTION TIME

To generate predictions on the subsampled test set (see above), we run EDC+ with the `Mistral-7B-Instruct-v0.2` and `gemma-7b` LLMs on two H100 GPUs for 24 hours.

### H.2 EDC+ PROMPTS

The following prompt is designed to identify all the operations to update the KG defined in Section 3. Concretely, it allows to identify triples explicitly mentioned in text under *Triples in text* category. This includes *X-Triples*, *E-Triples*, and *EE-Triples*. It also allows to classify *Triples in text* in those that should be added to the KG (i.e., with the ADD tag), and those that should be deprecated (i.e., with the DEPRECATE tag). This way, the prompt also facilitates the identification of KG triples that may need to be deprecated (i.e., *D-Triples*). Finally, the prompt allows to detect *EE-KG-Triples* under *Triples not in text* category, by asking LLM to identify triples with only one single entity (head or tail) mentioned in text, and the other entity existing in the KG.

```
Your task is to transform the given text into a semantic graph
in the form of a list of triples.  Two sets of triples are
to be extracted:  'Triples in text', which contain triples
relating entities mentioned in text in the form of [Entity1,
Relationship, Entity2, Action], where action indicates if the
triple has to be added (action 'ADD') or deprecated (action
'DEPRECATE') from the graph based on the knowledge in text.
The second set of triples is called 'Triples not in text', and
consists of triples with one entity (head or tail) mentioned in
text and the other entity not mentioned in text but potentially
existing in the graph.
In your answer, please strictly only include the triples and do
not include any explanation or apologies.
Here are some examples:

<FEW_SHOT_EXAMPLES>

Now please extract triplets from the following text.

Text:  <INPUT_TEXT>
```

## I RELIK EXPERIMENTAL CONFIGURATION

To generate predictions, we run ReLiK on each KG snapshot independently. In each run, ReLiK is provided with the dictionary of entities and relations specific to that snapshot. For relation encoding, we use the pre-trained ReLiK model available on Hugging Face: `relik-ie/encoder-e5-small-v2-wikipedia- relations`. These relation encodings are used by both ReLiK RE and ReLiK cIE. For each snapshot, we also encode the corresponding KG entities using the model `relik-ie/encoder-e5-small-v2-wikipedia-matryoshka`.

For prediction, we use the pre-trained `relik-ie/relik-relation-extraction-large` model for ReLiK RE, and the pre-trained `relik-ie/relik-cie-large` model for ReLiK cIE.

Running ReLiK on the subsampled EMERGE test set takes about 5 hours on a single A100 GPU.

## J WIKIDATA QUALIFIERS TO DETECT DEPRECATION OF TRIPLES

The following is the list of Wikidata qualifiers we use to detect the deprecation of triples when creating EMERGE:

1. P582: end time.

2. P1326: latest date.

3. P576: dissolved, abolished or demolished date.

4. P570: date of death.

5. P730: service retirement.

6. P2032: work period (end).

7. P2669: discontinued date.

8. P3999: date of official closure.

9. P7125: date of the latest one.

# K LIMITATIONS AND FUTURE WORK

In this work, we focus specifically on changes to the KG that reflect the introduction or modification of factual knowledge. We do not account for structural or curation-related changes that a KG may undergo, such as schema adjustments, property reorganization, or entity merging. These types of changes are often independent of new information appearing in external sources like Wikipedia and are typically driven by internal quality control or ontology refinement processes. While important for maintaining the integrity and usability of the KG, such changes fall outside the scope of our current study.

In this work, we focus on leveraging external textual sources to enhance KGs. However, textual data represents only one type of external knowledge. Other modalities—such as video (e.g., podcasts), images, and audio—also contain rich, complementary information that can contribute to KG enrichment. As such, a promising direction for future research is to explore the integration of knowledge from these multimodal sources to address this limitation.

During the generation of EMERGE, we use the same temporal delta window for both, the extraction of changes in Wikidata and the emerging passages from Wikipedia. However, certain pieces of knowledge do not always appear within the same time frame in the two sources. For example, events such as Brexit or the election of a president are often documented in Wikipedia months or even years before they are incorporated into the Wikidata knowledge graph. In future work, we plan to investigate this temporal discrepancy between the two knowledge sources more thoroughly.

Furthermore, this study restricts attention to triples in which both the subject and object are entities present in the entity catalog. Nonetheless, numerous valuable relations involve literals as objects, such as dates of birth, lengths, sizes, or employee counts (Mesquita et al., 2019), which are not considered in the current work.

Finally, a limitation of EMERGE is that it covers only the subset of Wikidata changes that can be reliably grounded in Wikipedia text. This stems from the fact that Wikidata is crowdsourced and not fully determined by Wikipedia content, meaning that many Wikidata updates have no corresponding textual evidence. In addition, EMERGE is restricted to Wikipedia paragraphs in which annotated entity mentions can be reliably identified through hyperlinks, as described in Section 4.1. As future work, an alternative dataset could be constructed using text-to-data generation methods (Hu et al., 2025; Edge et al., 2024; Hofer et al., 2024) to create a synthetic KG that mirrors all knowledge found in text, thereby achieving complete coverage of updates. While such an approach would ensure full alignment between text and KG, it would also introduce challenges such as potential errors in entity disambiguation and the substantial computational cost of generating an entire KG from text.

Another direction for future work is to incorporate explicit start and end dates for TKGU operations that imply changes in the KG, such as triple addition and deprecation. In the current version of EMERGE, deprecated facts are identified through the delta interval, as our work primarily focuses on updating the KG at a specific point in time rather than modeling full temporal validity. Adding explicit temporal qualifiers would more precisely capture when a fact begins and ceases to hold, aligning EMERGE more closely with the way temporal information is handled in Wikidata. This extension would also enable richer modeling of fact evolution and support downstream methods that rely on explicit temporal boundaries.

A further promising future direction is to develop information extraction methods that rely not only on textual evidence to extract triples, but also integrate this content with existing knowledge and patterns in the KG. Such an approach could be particularly beneficial for incorporating emerging entities in EE-KG-Triples, even when they are not supported by textual passages, into the broader KG.

## L  DATASET DOCUMENTATION: DATASHEET

We describe our dataset following the datasheets for datasets guidelines introduced in (Gebru et al., 2021), detailing its motivation, composition, collection process, and recommended uses. This documentation supports transparency, reproducibility, and responsible dataset use in machine learning research.

### L.1  MOTIVATION

**For what purpose was the dataset created?**   The EMERGE dataset was created to address the lack of integration between changes in textual knowledge and their effect on knowledge graph content. The proposed benchmark enables evaluation of KG updates driven by newly emerging knowledge in textual sources over temporally increasing KG deltas. Moreover, because the dataset is generated via an automatic annotation pipeline, it can be continuously extended to include more recent knowledge, thereby allowing evaluation of model robustness to ever-evolving and novel information and KG structures. This contrasts with existing benchmarks (see Table 4 in the Appendix A.1), which are static in nature and unable to emulate the evolution of knowledge in textual and KG sources (columns *Evolution-KG* and *Evolution-Text* in Table 4). Furthermore, existing benchmarks do not cover all the necessary text-driven knowledge graph update (TKGU) operations necessary to keep them updated (columns *X-Triples*, *E-Triples*, *EE-Triples*, *EE-KG-Triples* and *D-Triples* in Table 4).

We expect EMERGE will encourage the research on methods that are not limited to extracting knowledge from textual sources, but also capable of effectively maintaining KGs by integrating that knowledge into existing KGs. This contrasts with current state-of-the-art IE methods (see Section 2 and Table 1) limited to the extraction of knowledge purely from text without the ability to effectively integrate that knowledge into existing knowledge in KGs.

**Who created the dataset (e.g., which team, research group) and on behalf of which entity (e.g., company, institution, organization)?**   The dataset was developed by academic researchers through an international, cross-institutional collaboration. The contributing researchers bring extensive expertise in information extraction methods and dataset construction.

**Who funded the creation of the dataset?**   The dataset was created with funding from, among others, the highly prestigious European Union Marie Curie Actions Postdoctoral Grant.

### L.2  COMPOSITION

**What do the instances that comprise the dataset represent (e.g., documents, photos, people, countries)?**   The instances that comprise the dataset represent general-domain passages from Wikipedia, KG triples representing the knowledge contained in those passages, and TKGU operations (see Section 3) with respect to the respective general-domain Wikidata KG snapshot.

**How many instances are there in total (of each type, if appropriate)?**   Our EMERGE contains in total 233K instances, with a total of 1.4M TKGU operations: 727K X-Triples, 240K E-Triples, 161K EE-Triples, 313K EE-KG-Triples, and 8K D-Triples.

**Does the dataset contain all possible instances or is it a sample (not necessarily random) of instances from a larger set?**   We include a set with all possible instances that can be used for training. For testing (on which we report our results), we subsampled 1,000 instances per delta per snapshot.

**What data does each instance consist of?**   Each of the instances in the dataset consists of a textual passage with an annotated set of entity mentions linked to a particular KG snapshot. In addition, each instance includes a list of triples together with the corresponding TKGU operations that update the KG snapshot, as described in Section 3. Each triple is further annotated with an LLM-based assessment indicating whether the knowledge it represents can be inferred from the textual passage. See Appendix C.1 for details on the prompt and examples. The dataset spans seven yearly KG snapshots covering 2019-2025. For each snapshot, TKGU updates are annotated over five progressively larger weekly KG deltas, thereby capturing different levels of knowledge staleness in the KG.

**Is there a label or target associated with each instance?**   Yes, the target consists of all the triples with corresponding TKGU operations associated with the textual passage of an instance. These operations specify the updates to be applied to a KG snapshot to ensure consistency with the textual passage.

**Is any information missing from individual instances?**   All the instances are consistently annotated. However, the triples involved in TKGU operations associated with a passage are restricted to the entities of mentions explicitly annotated with hyperlinks in Wikipedia (see Section 4.1 for further details on annotation process). As such, there might be TKGU operations not covered by our dataset. This is also discussed in the limitations sections (see Section K).

**Are relationships between individual instances made explicit (e.g., users' movie ratings, social network links)?**   Yes, all the detected TKGU operations during the annotation process are made explicit. We further mark each of these operations as supported or no by the content of textual passage using LLM automatic annotation process described in Section 4.2.

**Are there recommended data splits (e.g., training, development/validation, testing)?**   Yes. We recommend training and validating models on earlier snapshots (e.g., from 2019 and 2020) and testing on later snapshots (i.e., from 2021–2025). This setup prevents knowledge leakage, since earlier KG snapshots do not contain information from later ones.

**Are there any errors, sources of noise, or redundancies in the dataset?**   We applied several quality-control measures, including removing duplicate or highly similar passages and filtering out passages with a low proportion of English words, among others described in Section 4.2. In addition, we manually annotated and verified a random subset of the dataset (see Section 4.2). Nevertheless, we do not consider EMERGE as entirely error-free, as it may contain factual inaccuracies resulting from erroneous edits in Wikipedia or Wikidata. Finally, the annotation agreement scores between the LLMs and human annotators, as well as between humans, are very strong (see Section Section 4.2) but not perfect, reflecting the complexity and intricacy of error detection in the dataset.

**Is the dataset self-contained, or does it link to or otherwise rely on external resources (e.g., websites, tweets, other datasets)?**   Yes, the introduced EMERGE dataset is self-contained and consists of:

1. Annotated instances containing passages with associated KG triples and TKGU operations.

2. Wikidata KG snapshots to which the annotated TKGU updates are applied.

**Does the dataset contain data that might be considered confidential (e.g., data that is protected by legal privilege or by doctor–patient confidentiality, data that includes the content of individuals' non-public communications)?**   No, Wikidata and Wikipedia are public resources.

**Does the dataset contain data that, if viewed directly, might be offensive, insulting, threatening, or might otherwise cause anxiety?**   No, no such instances were observed in EMERGE.

**Does the dataset identify any subpopulations (e.g., by age, gender)?**   While Wikipedia and Wikidata contain entities from various subpopulations, when building EMERGE, we do not focus on identifying and annotating any one in particular.

**Is it possible to identify individuals (that is, one or more natural persons), either directly or indirectly (that is, in combination with other data) from the dataset?** It is possible to identify individuals publicly described in Wikipedia pages or represented in Wikidata entities. However, we do not save other personal information, such as details of the editors involved in Wikipedia and Wikidata updates.

**Does the dataset contain data that might be considered sensitive in any way (e.g., data that reveals race or ethnic origins, sexual orientations, religious beliefs, political opinions or union memberships, or locations; financial or health data; biometric or genetic data; forms of government identification, such as social security numbers; criminal history)?** Since Wikipedia and Wikidata are public resources intended to be factual, this concern can be disregarded for the majority of instances in EMERGE.

## L.3 COLLECTION PROCESS

**How was the data associated with each instance acquired? Was the data directly observable (e.g., raw text, movie ratings), reported by subjects (e.g., survey responses), or indirectly inferred/derived from other data (e.g., part-of-speech tags, model-based guesses for age or language)?** The EMERGE dataset was annotated using publicly available entity mentions in Wikipedia pages, as described in Section 4.1. These hyperlinked mentions are visible to any Wikipedia visitor as links to other pages. To annotate the TKGU operations, we relied on actual updates in Wikidata. Generative models (i.e., LLMs) were used only to verify whether the detected TKGU operations are reflected in the textual content of the passages (see Section 4.2).

**What mechanisms or procedures were used to collect the data (e.g., hardware apparatuses or sensors, manual human curation, software programs, software APIs)?** The EMERGE dataset was generated from the Wikipedia and Wikidata dumps of March 2025. A computing cluster with 64 CPUs and 128 GB of RAM was used to process and generate the dataset. Additionally, a cluster with 4 H100 GPUs was used to run `Meta-Llama-3.1-405B` for verifying that the TKGU operations are effectively represented in the textual passages (see Section 4.2).

**If the dataset is a sample from a larger set, what was the sampling strategy (e.g., deterministic, probabilistic with specific sampling probabilities)?** The test set used in our experiments was randomly sampled from the larger dataset, with a maximum of 1,000 instances per snapshot per KG delta. The sampling procedure, described in detail in Section Section 5, includes retention of a minimum of 400 instances per delta for operations that require actual updates to the KG (i.e., D-Triples, E-Triples, EE-Triples, and EE-KG-Triples). This ensures that the models are evaluated on a sufficiently large number of such instances. This is particularly important for D-Triples TKGU operations, which are very scarce in the original dataset; without this retention, a purely random subsample would contain only a few instances, potentially leading to high variability in the results.

**Who was involved in the data collection process (e.g., students, crowdworkers, contractors) and how were they compensated (e.g., how much were crowdworkers paid)?** The dataset was generated automatically from real-world updates to Wikidata and changes in Wikipedia articles. LLMs were used to assess each TKGU operation with respect to the knowledge contained in the textual passages. The only human involvement was the annotation of a subsample of the dataset to measure agreement with the LLM annotations. For this purpose, two researchers acted as annotators and were credited as co-authors of the paper.

**Over what timeframe was the data collected?** The data were collected from seven yearly snapshots, spanning January 1, 2019, to January 1, 2025. For each snapshot, KG deltas were extracted for up to five weeks, ending on February 5 of the corresponding year.

**Were any ethical review processes conducted (e.g., by an institutional review board)?** No, the public nature of the data, consisting of Wikipedia pages and Wikidata KG updates, meant that no formal ethical review was required.

**Did you collect the data from the individuals in question directly, or obtain it via third parties or other sources (e.g., websites)?**   The data were obtained from publicly available Wikipedia and Wikidata repository dumps (`https://dumps.wikimedia.org/`).

**Were the individuals in question notified about the data collection?**   No individuals were directly involved in the data collection.

**Did the individuals in question consent to the collection and use of their data?**   No individuals were directly involved in the data collection.

**If consent was obtained, were the consenting individuals provided with a mechanism to revoke their consent in the future or for certain uses?**   No individuals were directly involved in the data collection.

**Has an analysis of the potential impact of the dataset and its use on data subjects (e.g., a data protection impact analysis) been conducted?**   No formal data protection impact analysis was conducted, as the dataset is derived entirely from publicly available Wikipedia pages and Wikidata KG updates and does not include private or sensitive information about individuals.

L.4   PREPROCESSING/CLEANING/LABELING

**Was any preprocessing/cleaning/labeling of the data done (e.g., discretization or bucketing, tokenization, part-of-speech tagging, SIFT feature extraction, removal of instances, processing of missing values)?**   Yes. The original raw data from the Wikipedia and Wikidata dumps underwent several preprocessing steps:

1. Preprocessed Wikipedia wikitext, retaining only lists and textual paragraphs as dataset inputs, while excluding tables, figures, and other multimodal elements.

2. Extracted only Wikipedia text containing explicitly annotated entity mentions by editors, which could be mapped to Wikidata updates within a given time window in the KG delta.

3. Constrained Wikipedia passages to lengths between 30 and 1,000 tokens.

4. Filtered out passages with fewer than 30% English words, using the Python `nltk` package.

5. Applied stability constraints by discarding changes in Wikidata and Wikipedia that were quickly rolled back (often indicating incorrect knowledge). Specifically, we retained Wikidata KG updates persisting at least 7 days and Wikipedia edits not followed by another change within 30 minutes.

6. Ensured diversity by requiring passages aligned to similar updates in Wikipedia to differ in content, measured by edit distance (minimum 0.15 for texts under 2,500 characters and 0.25 for texts 2,500 characters or longer).

7. Validated the alignment of TKGU operations to textual passages with LLMs, explicitly marking operations that could be grounded in the passage content (see Section 4.2 for further details).

**Was the "raw" data saved in addition to the preprocessed/cleaned/labeled data (e.g., to support unanticipated future uses)?**   Yes. We preserved all input and output data from each preprocessing step, beginning with the raw Wikipedia and Wikidata dumps used to construct EMERGE.

**Is the software that was used to preprocess/clean/label the data available?**   Yes, all the software that was used to preprocess/clean/label will be publicly released upon acceptance.

L.5   USES

**Has the dataset been used for any tasks already?**   Yes, in Section 5 we experiment with various current state-of-the-art information extraction models.

**Is there a repository that links to any or all papers or systems that use the dataset?** Yes, there is a repository (currently private due to anonymity policy), which will be made public upon acceptance.

**What (other) tasks could the dataset be used for?** Beyond the KG updating task presented in this paper, EMERGE could be directly applied to at least the following tasks:

1. Question answering over novel and emerging knowledge derived from the TKGU operations introduced here.

2. General knowledge graph completion, where certain changes may trigger additional updates that are not limited to entities mentioned in textual passages but instead depend on the evolving KG structure. To support this, we will release all KG changes, not only those aligned with textual passages, which form the core of EMERGE.

**Is there anything about the composition of the dataset or the way it was collected and preprocessed/cleaned/labeled that might impact future uses?** No.

**Are there tasks for which the dataset should not be used?** No.

L.6 DISTRIBUTION

**Will the dataset be distributed to third parties outside of the entity (e.g., company, institution, organization) on behalf of which the dataset was created?** Yes, the dataset will be made publicly available in Hugging Face.

**How will the dataset be distributed (e.g., tarball on website, API, GitHub)?** The EMERGE dataset will be distributed via Hugging Face (`https://huggingface.co/`), and the code for generating the dataset will be released on GitHub (`https://github.com/`).

**When will the dataset be distributed?** The EMERGE dataset will be released publicly upon acceptance of the paper.

**Will the dataset be distributed under a copyright or other intellectual property (IP) license, and/or under applicable terms of use (ToU)?** To support openness and collaboration in research, we release the datasets under the Creative Commons Attribution 4.0 International (CC BY 4.0) license. The full terms of this license can be found on the Creative Commons website: `https://creativecommons.org/licenses/by/4.0/`.

**Have any third parties imposed IP-based or other restrictions on the data associated with the instances?** No, the dataset is derived from publicly available Wikipedia and Wikidata knowledge repositories and is not subject to any third-party IP restrictions.

**Do any export controls or other regulatory restrictions apply to the dataset or to individual instances?** No, the dataset and its individual instances are based on publicly available Wikipedia and Wikidata content and are not subject to export controls or other regulatory restrictions.

L.7 MAINTENANCE

**Who will be supporting/hosting/maintaining the dataset?** The dataset will be supported, hosted, and maintained by the authors of this paper.

**How can the owner/curator/manager of the dataset be contacted (e.g., email address)?** The dataset is curated and managed by the authors of this paper. Inquiries regarding the dataset, including access, usage, and reporting issues, can be directed to the corresponding authors via email. Additionally, users can submit questions or report issues through the GitHub repository hosting the dataset generation code.

**Is there an erratum?**    No erratum has been issued for the EMERGE dataset.

**Will the dataset be updated (e.g., to correct labeling errors, add new instances, delete instances)?**    Yes, EMERGE will be regularly updated with emerging knowledge through yearly snapshots. Announcements regarding new versions will be communicated via the EMERGE GitHub repository. Additionally, as described in Section 4.4, users can generate customized versions of EMERGE by adjusting relevant hyperparameters, as well as personalized snapshots of different granularity (e.g., daily, weekly, monthly).

**If the dataset relates to people, are there applicable limits on the retention of the data associated with the instances (e.g., were the individuals in question told that their data would be retained for a fixed period of time and then deleted)?**    The EMERGE dataset does not contain private or personally identifiable information about individuals. It is derived entirely from publicly available Wikipedia pages and Wikidata entities, and no retention limits for individual consent were applicable.

**Will older versions of the dataset continue to be supported/hosted/maintained?**    Yes, all previous versions of EMERGE will continue to be supported, hosted, and maintained. Each version will be assigned a unique version number, and we will provide persistent links to access every version through Hugging Face storage server. This will ensure reproducibility of experiments and will enable users to reference or use specific dataset versions as needed.

**If others want to extend/augment/build on/contribute to the dataset, is there a mechanism for them to do so?**    Yes. As described in Section 4.4, EMERGE users will have access to all necessary scripts to re-generate the dataset with customized settings. This includes adjusting hyperparameters such as the maximum passage length, generating the dataset for newer snapshots, and specifying the number and granularity of KG deltas.

# M    ACCESSIBILITY

The EMERGE will be released publicly via a Hugging Face repository. The accompanying code for extending it with emerging Wikipedia and Wikidata knowledge will be made available in a public GitHub repository. In addition, the test set used in our experiments is included as supplementary material with this submission.

