# OpenReview forum: "EMERGE: A Benchmark for Updating Knowledge Graphs with Emerging Textual Knowledge"
_ICLR.cc/2026/Conference — Submitted to ICLR 2026_

### Official Review · Reviewer_pLRm · 2025-10-15

**Soundness:** 2
**Presentation:** 2
**Contribution:** 1
**Rating:** 2
**Confidence:** 5

**Summary:**

This paper introduces EMERGE, a large-scale benchmark designed for studying how knowledge graphs can be automatically updated with new information from unstructured text sources such as Wikipedia. Unlike most static datasets, EMERGE aligns evolving Wikipedia passages with temporal Wikidata snapshots, producing over 233K passages and 1.45 million text-driven KG update operations. These operations include adding, modifying, and deprecating triples to reflect newly emerging knowledge.

**Strengths:**

1.	Comprehensive related works are mentioned to position it within the broader research landscape including knowledge graph completion, refinement, and information extraction.

2.	The experimental settings and evaluation details are provided, allowing for clearly understanding of the resource.

3.	The paper is generally clearly-written and easy to follow.

**Weaknesses:**

1.	The practical utility of the resource remains unclear. More concrete use cases, applications, or empirical demonstrations are needed to justify its real-world usefulness.

2.	The work focuses solely on factual updates (new or deprecated triples) while not considering schema-level or structural modifications, such as ontology changes, entity merges, or property reorganizations.

3.	The dataset is limited to entities presented in Wikidata while omitting literal values (e.g., numerical or date attributes), constraining its applicability to many factual domains.

4.	The benchmark evaluation only reports recall/completeness without precision or F1, which reduces the comprehensiveness of performance assessment under open-world assumptions.

5.	The dataset assumes synchronized temporal windows between textual and KG updates, however in practice, Wikipedia and Wikidata could diverge significantly in update timing.

**Questions:**

1.	How can this resource be effectively leveraged by other research communities or real-world applications? In what ways does it offer measurable advantages over existing datasets?

2.	What is the unique benefit of EMERGE compared to directly utilizing sequential Wikidata snapshots for temporal KG analysis?

---

> ### Author Response · Authors · 2025-11-24
> **Response to the reviewer's identified weaknesses**
>
> __1. On the practical utility, concrete use cases, applications, and real-world usefulness.__
>
> We thank the reviewer for raising this important point. EMERGE enables a task that existing IE resources aligned with KGs (such as REBEL, T-REX, and others in Table 4) do not support: text-driven KG updating (TKGU) conditioned on the current KG state. This includes adding emerging entities, updating relations between new and existing nodes, and deprecating outdated or contradicted triples driven by emerging knowledge in external textual knowledge sources. These capabilities are essential for downstream applications such as question answering, search assistants, knowledge-augmented LLMs, digital archives, and news-driven knowledge monitoring systems, all of which rely on up-to-date and text-verifiable KG content. We have highlighted this by re-strutured the introduction of our work, specifically in the first paragraph (lines 36–49) of the new introduction section.
>
> ---
>
> __2. On schema-level or structural modifications.__
>
> We understand the reviewer’s concern. In EMERGE we do account for KG structural modifications by constructing the dataset using 7 different yearly Wikidata snapshots (see Section 4.1). Each snapshot differs structurally—for example, in the set of entities and in the set of relation types (i.e., Wikidata properties)—as showcased in the “KG statistics” group of Table 6 in Appendix D.1. These structural differences are naturally reflected in the TKGU operations produced from each snapshot. Accordingly, we envision models that are robust to such KG changes and able to adapt to any of the snapshots included in EMERGE.
>
> ---
>
> __3. On omitting literal values.__
>
> We acknowledge the reviewer’s point. As noted in the limitations section (lines 2247–2250 of the revised manuscript), EMERGE currently focuses on entity–entity relations and does not include literal values such as dates or numerical attributes. Capturing such literals poses additional challenges, as they are not consistently annotated in Wikipedia text and are therefore difficult to extract and reliably align with the corresponding literal values in Wikidata. While extending EMERGE to support literal-based updates is a valuable direction for future work, we believe the current version remains highly useful, as it provides the first large-scale resource for studying text-driven KG updating in a naturally evolving text–KG setting.
>
> ---
>
> __4. On only reporting recall/completeness without precision or F1.__
>
> We appreciate the reviewer’s concern. We do not report precision or F1 in the main results (Tables 2–3) because these metrics can be misleading under the open-world assumption (Razniewski et al., 2024), where correct predictions may be mislabeled as false positives due to KG incompleteness. __Nevertheless, in response to the reviewer’s comment, we now include precision and F1 scores in Tables 13–16 of Appendix F.1 in the revised manuscript.__
>
> ---
>
> __5. On the use of temporal windows while Wikipedia and Wikidata could diverge significantly in update timing.__
>
> We agree that Wikipedia and Wikidata are not perfectly synchronized and may differ in update timing. We explicitly acknowledge this in the limitations section (lines 2241–2246 of the revised manuscript). In EMERGE, we approximate alignment by using fixed temporal windows between Wikidata snapshots, and this window is a configurable hyperparameter for users who regenerate the dataset. Our goal is not strict temporal simultaneity, but a reasonable, adjustable range for identifying passages that support KG updates.

---

> ### Author Response · Authors · 2025-11-24
> **Response to the reviewer’s raised questions**
>
> __Q1. How can this resource be effectively leveraged by other research communities or real-world applications? In what ways does it offer measurable advantages over existing datasets?__
>
> EMERGE is the first benchmark designed specifically for text-driven KG updating (TKGU), a capability not supported by existing IE datasets. Its methodology—snapshot construction, text–KG alignment, and explicit TKGU operation labeling—can be applied to dynamic domains such as biomedical, financial, legal, and scientific KGs. This directly benefits applications that require timely KG updates, including QA, search, retrieval-augmented LMs, and event monitoring.
>
> EMERGE provides several measurable advantages:
>
> _1. Explicit formalization of KG update operations._
>
> It is the first dataset to define and annotate the full set of TKGU operations (E-, EE-, EE-KG-, D-, X-Triples). Existing IE datasets extract triples but do not map them to concrete update actions, preventing tasks like identifying deprecated triples or adding/connecting emerging entities.
>
> _2. Enabling KG-aware IE models._
>
> By exposing gaps in current IE systems—e.g., handling deprecations, reasoning over KG structure, or linking new entities—EMERGE supports the development of models that jointly use text and KG context.
>
> _3. Realistic temporal knowledge evolution._
>
> EMERGE grounds updates in real Wikidata and Wikipedia snapshots, capturing how knowledge naturally changes over time. Prior IE datasets (e.g., REBEL, T-REX) lack this temporal and structural grounding.
>
> A detailed comparison appears in Table 4 (Appendix A). Additionally, the Introduction (lines 36–49) has been updated to highlight EMERGE’s real-world relevance.
>
> ---
>
> __Q2. What is the unique benefit of EMERGE compared to directly utilizing sequential Wikidata snapshots for temporal KG analysis?__
>
> EMERGE provides benefits that go beyond directly using sequential Wikidata snapshots. While snapshots show how the KG changes over time, they do not reveal how these changes are grounded in textual evidence or derived from external sources such as Wikipedia. EMERGE addresses this by aligning a subset of KG updates with the specific Wikipedia passages that support them, enabling the study of text-driven KG updating rather than temporal evolution alone. This allows evaluation of models on update operations grounded in external text – such as adding emerging entities, updating relations supported by passages, and identifying text-supported deprecations – which snapshots by themselves cannot provide.
>
> To clarify this distinction, we have added a discussion contrasting temporal KG completion with our setting in lines 138–148 of the related work section, and included the discussion of KG versioning in the extended related work in Appendix A.2 (lines 1350–1358).

---

### Official Review · Reviewer_V2ha · 2025-11-01

**Soundness:** 3
**Presentation:** 2
**Contribution:** 3
**Rating:** 6
**Confidence:** 4

**Summary:**

This paper addresses the problem of how to correctly map newly emerging textual facts to the required updates of a knowledge graph (KG) at a specific point in time. The authors formalize the Text-driven Knowledge Graph Updating (TKGU) task and define five update operation types: X-Triples, E-Triples, EE-Triples, EE-KG-Triples, and D-Triples. They further construct EMERGE, a large-scale and incrementally extensible benchmark aligned between Wikipedia text and weekly deltas of Wikidata, spanning 2019–2025 with 233K instances and ~1.45M KG update operations. Using two families of advanced information extraction baselines, they conduct systematic evaluation across multiple snapshots and temporal deltas, showing that existing IE/LLM methods struggle with connecting emerging entities back to the KG and with fact revocation operations. In addition, performance degradation increases with larger deltas, indicating that EMERGE reveals realistic capability gaps in dynamic KG maintenance.

**Strengths:**

S1: The paper not only proposes the direction of “text-driven KG updating” but also formalizes the task into five concrete TKGU operations (Section 3). With clear examples (Figure 1), the classification criteria are explicit and allow direct validation and quantification, going beyond generic triple extraction.

S2: EMERGE covers seven yearly snapshots (2019–2025) and up to five-week cumulative deltas per snapshot, totaling 233K instances and ~1.45M updates, with significant growth in KG entities and relations over time. This provides a solid foundation for studying robustness to temporal evolution and schema drift.

S3: The benchmark construction leverages Wikipedia/Wikidata historical dumps and includes weekly snapshot/delta generation, text–KG alignment, and data curation pipelines. Cleaning rules and rollback filtering strategies are documented, supporting extensibility and reproducibility.

S4: The evaluation is detailed across different models, snapshots, and deltas using completeness/recall metrics (Table 2, Figure 3, Appendix). Qualitative failure examples (Appendix E) help identify future research directions (e.g., the need for models to exploit KG structure rather than text alone).

**Weaknesses:**

W1: Although Meta-Llama-3.1 is used to filter alignment pairs efficiently, details of the LLM filtering process—thresholds, prompt designs, and rules for marking updates as “unsupported”—are insufficiently provided, affecting interpretability and reproducibility of the dataset.

W2: D-Triples constitute only 0.6% of the full dataset (3.3% in the subsampled test set), and their correctness often relies on KG-level reasoning. The imbalance and uncertainty make the evaluation unstable, and the paper does not demonstrate mitigation strategies for potential misjudgment or skewed metrics.

W3: EE-KG-Triples rely on external KG knowledge or schema assumptions (e.g., typing a new entity as Human). Many such links are not textually supported, yet the paper does not provide a clear evaluation protocol or manual validation rate to confirm their accuracy.

W4: Closed-IE ReLiK models and open-generation EDC+ differ significantly in the amount of KG information provided (entity/relation dictionaries vs. prompt-based access). The comparison risks conflating model capabilities with differences in provided prior knowledge.

W5: The definitions of EE-KG and D-Triples are introduced without intuitive examples in the main paper, leaving readers dependent on the Appendix for clarity.

W6: Section 4 (dataset construction) is disproportionately long, while Section 5 (analysis) offers limited error type exploration and insufficient discussion of challenge sources.

**Questions:**

Q1: How are the non-textually supported EE-KG links retained (fully Wikidata-driven?), and can the authors provide manual evaluation results (accuracy, common error types) for a small EE-KG sample?

Q2: The authors state that code/data will be released after acceptance, but even in the anonymized version, please provide: data format specification (JSON schema), alignment output examples.

Q3: Could the authors clarify whether D-Triples rely strictly on explicit deletion operations or also semantic overrides? Please also provide Cohen’s κ / F1 from human annotation if available.

---

> ### Author Response · Authors · 2025-11-24
> **Response to the reviewer's identified weaknesses**
>
> __W1: Although Meta-Llama-3.1 is used to filter alignment pairs efficiently, details of the LLM filtering process - thresholds, prompt designs, and rules for marking updates as “unsupported” - are insufficiently provided, affecting interpretability and reproducibility of the dataset.__
>
> We understand the reviewer’s concern. Full details of our LLM filtering process, including prompts and illustrative examples, are provided in Appendix C.1. We have added an explicit reference to this appendix in lines 299-300 of the revised manuscript to improve clarity and reproducibility.
>
> ---
>
> __W2: On D-triple imbalance and evaluation instability__
>
> We understand the reviewer’s concern regarding the small proportion of D-Triples. While D-Triples represent a minority of the dataset, this does not affect metric stability because we evaluate each TKGU operation type independently rather than aggregating performance across operation types. We have clarified this point in lines 343-345 of the revised manuscript.
>
> ---
>
> __W3: EE-KG-Triples rely on external KG knowledge or schema assumptions (e.g., typing a new entity as Human). Many such links are not textually supported, yet the paper does not provide a clear evaluation protocol or manual validation rate to confirm their accuracy.__
>
> We understand the reviewer’s concern regarding the textual support for EE-KG-Triples. The manuscript reports detailed statistics on how many EE-KG-Triples are marked as text-supported by the Meta-Llama-3.1-405B annotator model (see Appendix C.3, referenced in lines 304–306 of the revised manuscript). As the reviewer anticipates, only a minority of EE-KG-Triples (36% in the test set) is flagged as supported by the text (see Figure 6), which is substantially lower than the 73–86% support rates observed for the other TKGU operation types. To evaluate annotation agreement between humans and the LLM-based filtering, we conduct human evaluation on a random subsample of all TKGU operations and compare it with Meta-Llama-3.1-405B assessments (see lines 307–311 of Section 4.2 and Appendices C.2.1–C.2.2 of the revised manuscript). Agreement for EE-KG-Triples remains strong for both annotators, with Cohen’s kappa scores above 0.75 as reported in Table 5 of the appendix, providing further validation of annotation quality.
>
> ---
>
> __W4: Closed-IE ReLiK models and open-generation EDC+ differ significantly in the amount of KG information provided (entity/relation dictionaries vs. prompt-based access). The comparison risks conflating model capabilities with differences in provided prior knowledge.__
>
> We appreciate the reviewer’s observation. Our aim is not to compare the absolute performance of ReLiK cIE and EDC+, as they differ in how much KG information they access (dictionary-based lookup vs. prompt-based generation). Rather, we use them to illustrate the complementary limitations of current IE paradigms when applied to text-driven KG updating. The results show that neither architecture can reliably handle all TKGU operation types or effectively integrate and reason over KG content. In particular, ReLiK cIE cannot address EE-KG-Triples or detect D-Triples without architectural changes, while EDC+ can only partially cover such cases through prompting, with low performance. This highlights the need for future methods that interact more directly with KG structure. We have clarified this intent and highlighted these limitations in lines 354-359 of the revised manuscript.
>
> ---
>
> __W5: The definitions of EE-KG and D-Triples are introduced without intuitive examples in the main paper, leaving readers dependent on the Appendix for clarity.__
>
> We appreciate the reviewer’s comment. Intuitive examples of all TKGU operation types, including EE-KG-Triples and D-Triples, are already provided in Section 3, and correspond directly to the dataset instance in Figure 1; these cases are described in lines 211–219 of the revised manuscript, and linked to concrete examples in the figure. Additional illustrative examples for every TKGU operation type are included in Tables 7–12 in Appendix E. We have also added explicit references to these tables in line 323 of the revised manuscript.
>
> ---
>
> __W6: Section 4 (dataset construction) is disproportionately long, while Section 5 (analysis) offers limited error type exploration and insufficient discussion of challenge sources.__
>
> Thank you for pointing this out. In response, we have expanded the analysis in Section 5 to more closely examine the sources of decreased performance as the deltas increase. In particular, we now discuss the performance drop for larger deltas (lines 509–522), showing that the increasing number of relation types at higher deltas is one contributing factor. To support this finding, we have also incorporated Figures 4 and 5.

---

> ### Author Response · Authors · 2025-11-24
> **Response to the reviewer’s raised questions**
>
> __Q1: How are the non-textually supported EE-KG links retained (fully Wikidata-driven?), and can the authors provide manual evaluation results (accuracy, common error types) for a small EE-KG sample?__
>
> As discussed in Section 4.2 (lines 300–301 of the revised manuscript), all aligned TKGU operation triples, including EE-KG triples, are retained in EMERGE. Triples that the Meta-Llama-3.1-405B annotator judges as not textually supported are not removed but explicitly flagged as unsupported, allowing future re-verification with more capable LLMs. In our experiments, however, only triples flagged as supported are counted during evaluation (e.g., in Tables 2 and 3).
>
> Regarding manual evaluation, as noted in our response to W3, we report human–LLM annotation agreement in Table 5 of Appendix C.2.2, and provide detailed statistics on supported versus unsupported triples for each TKGU operation type in Figure 6. In addition, we now include a table in Appendix E (Table 11; see also lines 1745–1751) summarizing the most common EE-KG triples that were marked as unsupported by the passage by the LLM in the test set.
>
> -----
>
> __Q2: The authors state that code/data will be released after acceptance, but even in the anonymized version, please provide: data format specification (JSON schema), alignment output examples.__
>
> Thank you for the suggestion. In addition to our commitment to release the full dataset and code upon acceptance, we have uploaded the test set used in our experiments, along with the manually annotated subset used to report human–LLM agreement (Appendix C.2), as a supplementary material zip file.
>
> -----
>
> __Q3: Could the authors clarify whether D-Triples rely strictly on explicit deletion operations or also semantic overrides? Please also provide Cohen’s κ / F1 from human annotation if available.__
>
> D-Triples in EMERGE capture all forms of triple deprecation reflected in the Wikidata revision history. This includes both explicit deletion operations and cases where a triple becomes invalid due to a semantic override (i.e., a new, conflicting fact replaces the old one). Because we track the full temporal history of each triple, both types of deprecations appear naturally in our deltas. However, EMERGE does not explicitly distinguish between explicit deletions and semantic overrides; the latter is implicitly represented when a passage supports the deprecation of an existing triple and, if applicable, the creation of a new one. If the passage only supports the deprecation (without supporting a new contradictory fact), then only the deprecated triple is recorded.
> Regarding annotation quality, we provide Cohen’s κ statistics for human–human and human–LLM agreement in Appendix C.2.2, Table 5 on a subsample of dataset. We additionally report multi-rater agreement across both human annotators and the LLM using Fleiss’ κ and Krippendorff’s alpha.

---

### Official Review · Reviewer_wRHi · 2025-11-02

**Soundness:** 1
**Presentation:** 1
**Contribution:** 1
**Rating:** 0
**Confidence:** 5

**Summary:**

The paper proposes a pipeline as well as a benchmark dataset for the updates in the knowledge graphs.

**Strengths:**

--> The pipeline for automated updates from text could be a useful tool to integrate for the live updates of the major general purpose knowledge graphs such as Wikidata or DBpedia.

**Weaknesses:**

- In the abstract the authors could give a bit of the introduction of how is the framework implemented.
- The interchangeable use of the terms knowledge graphs and knowledge bases is quite well known, it does not need to be explicitly said in footnote 2.
- There are many other datasets available for KG completion such as LiterallyWikidata [1] or Codex [2], etc.
- The authors are discussing the SoTA on KG Completion but this task is not discussed later on or the framework is not evaluated on that task. Why are the authors relating this task to that family of algorithms?
- Why is the SoTA on information extraction is discussed since this is also not the main focus, the main focus is on how to handle updates in the knowledge graph.
- the authors should also discuss the methods for ontology versioning, etc. in the SoTA and make a clear distinction [3].
- The authors should keep in mind that Wikidata is also crowdsourced on top of containing the information from Wikipedia.
- The part on "Emerging entities to KG triples..." discusses about "instanceOf" relation which is a different problem, i.e., entity typing and adding new instances to an entity.
- The example of deprecated triple seems misleading. If a person stops being a member of a club then there should be added an end date because the fact doesn't change it just expires overtime.

[1] https://link.springer.com/chapter/10.1007/978-3-030-88361-4_30
[2] https://arxiv.org/pdf/2009.07810
[3] https://arxiv.org/pdf/2409.04572

**Questions:**

- The authors are discussing the SoTA on KG Completion but this task is not discussed later on or the framework is not evaluated on that task. Why are the authors relating this task to that family of algorithms?
- Why is the SoTA on information extraction is discussed since this is also not the main focus, the main focus is on how to handle updates in the knowledge graph? It should talk about Entity Linking algorithms and the existing dataset which suffer from the updates in the KG.
- Overall, the problem statement of the paper is a bit confusing. The pipeline is for updates in the knowledge graph with a dataset which enriches the Knowledge Graph with updated information but the evaluation seems to be on entity linking which is another problem linked with updated knowledge graphs and the text on which the entity linking is performed.
- the paper might be a better fit for the evaluations on dynamic knowledge graph completion.

---

> ### Author Response · Authors · 2025-11-24
> **Response to the reviewer's identified weaknesses**
>
> Thank you for your detailed comments. Could you please clarify what specifically led you to assign the lowest possible score to our manuscript, beyond the listed weaknesses, so we can better understand the overall assessment and improve the work accordingly?
>
> __1. On the limited technical description in the abstract.__
>
> We have added lines 22–25 with extra implementation details.
>
> ---
>
> __2. On unnecessary footnote on KG/KB equivalence.__
>
> Footnote removed in the revised manuscript.
>
> ---
>
> __3. On additional KGC datasets such as LiterallyWikidata or Codex , etc.__
>
> Thank you for the suggestion. The CoDEx reference was already included, and we have now added the recommended LiterallyWikidata citation in line 138. To maintain a clear focus, we emphasize Wikidata-based (or similar, such as Freebase and YAGO) datasets, which are most closely aligned with our contribution and benchmark design.
>
> ---
>
> __4. On relating our work to the SoTA on KG Completion.__
>
> We appreciate the reviewer’s concern. We discussed existing KG completion datasets because they are commonly used to train and evaluate models, but they focus solely on predicting new edges within the KG’s internal structure. Our work instead targets KG updates driven by information extracted from external unstructured text. We clarified this distinction in lines 138–141 and also added the relevant Temporal KG Completion references in lines 141–148, including the reviewer’s suggested citation.
>
> ---
>
> __5. On relating our work to the SoTA on information extraction (IE).__
>
> We understand the reviewer’s concern. Existing IE datasets link text to KG triples but do not model the update operations required when new information emerges (see detailed comparison with EMERGE in Table 4). Our work addresses this gap by mapping textual knowledge to concrete KG update operations and showing that current IE methods cover only part of them (Table 1). We benchmark these methods because they provide the basis for future IE models that can combine textual signals with KG structure to support all TKGU operations. We clarified this in the introduction (lines 85–95) and the related work section (lines 187–192) and outlined the necessary extensions in our future work discussion (lines 2268–2272) of the newly uploaded manuscript version.
>
> ---
>
> __6. On adding discussion and positioning against ontology versioning methods .__
>
> Our work does perform KG versioning when creating yearly snapshots and deltas from Wikidata. However, unlike approaches that focus on maintaining or comparing temporal KG states, we use versioning only as an intermediate step to build a dataset where each KG update is linked to its supporting Wikipedia evidence. Thus, EMERGE is not a versioned KG, but a text-grounded IE benchmark that maps textual passages to the KG updates they induce. We added the suggested reference and clarified this distinction in the revised manuscript (lines 1350–1358) of the extended related work section in Appendix A.2.
>
> ---
>
> __7. On the crowdsourced nature of Wikidata.__
>
> We agree that Wikidata is crowdsourced and not fully determined by Wikipedia content. EMERGE therefore does not aim to capture all Wikidata changes, but only the subset of updates that can be reliably grounded in Wikipedia through our alignment pipeline (Section 4). This yields 1.45M KG edits aligned with 233K passages, which we consider a representative and useful scale for IE model training and evaluation. We clarified this in the limitations and future work section (lines 2252–2260) and briefly discussed an alternative text-to-data generation approach.
>
> ---
>
> __8. On "instanceOf" relation.__
>
> We appreciate the reviewer’s comment. While instanceOf is indeed often associated with entity typing, in Wikidata it is simply one of many relation types defined in the schema and is treated like any other edge. EMERGE therefore includes instanceOf updates in the same way as all other KG triples, and such updates can appear in any TKGU operation when supported by the text. Our focus is not on entity typing itself, but on grounding all text-induced KG updates across the full Wikidata schema.
>
> ---
>
> __9. On triple deprecation and adding end date to expired facts.__
>
> We agree that facts such as membership should ideally be marked as expired rather than removed. In EMERGE, we follow this principle: deprecated triples are not deleted but flagged as no longer valid at a given snapshot. The current version does not yet assign explicit end dates, as our focus is on keeping the KG up to date with text at each snapshot and its delta (Section 4). We clarified that D-Triples indicate deprecation rather than deletion (lines 277–279) and noted that adding explicit start and end dates is a natural direction for future work (lines 2261–2267) in the revised manuscript version.

---

> ### Author Response · Authors · 2025-11-24
> **Response to the reviewer’s raised questions**
>
> __1. The authors are discussing the SoTA on KG Completion but this task is not discussed later on or the framework is not evaluated on that task. Why are the authors relating this task to that family of algorithms?__
>
> See answer to weakness 4 in the “Response to the reviewer's identified weaknesses” comment.
>
> ---
>
> __2. Why is the SoTA on information extraction is discussed since this is also not the main focus, the main focus is on how to handle updates in the knowledge graph? It should talk about Entity Linking algorithms and the existing dataset which suffer from the updates in the KG.__
>
> See answer to weakness 5 in the “Response to the reviewer's identified weaknesses” comment.
>
> ---
>
> __3. Overall, the problem statement of the paper is a bit confusing. The pipeline is for updates in the knowledge graph with a dataset which enriches the Knowledge Graph with updated information but the evaluation seems to be on entity linking which is another problem linked with updated knowledge graphs and the text on which the entity linking is performed.__
>
> We understand the reviewer’s concern regarding the problem statement and evaluation setup. Our goal is to evaluate models on their ability to predict text-driven KG update (TKGU) operations in EMERGE. For close information extraction (IE) models, such as ReLiK cIE, which we use for E-Triples and X-Triples tasks (see results in Table 3), evaluation is straightforward because these models link extracted triples to the KG (i.e., to entity and relation IDs), allowing exact matching with annotated updates. However, some TKGU operations (e.g., EE-Triples and EE-KG-Triples) involve entities not yet present in the KG, making ID-based matching impossible. IE models like ReLiK RE and EDC+ can extract such triples but do not link them to the KG, so we evaluate them using the completeness score, which compares textual triples via cosine similarity (see Appendix B.2). Current IE models also lack awareness of existing KG content and therefore cannot identify deprecated triples (D-Triples). For this reason, we use an adapted IE EDC+ model to extract triples that may need to be deprecated based solely on text, which again requires completeness-based evaluation. This reflects a broader gap: existing IE models operate independently of KG structure, limiting their ability to perform true text-driven KG updating. We see promising future directions in developing IE methods that jointly exploit textual evidence and KG state to generate grounded TKGU operations, as discussed in the limitations and future work section (lines 2268-2272). We have also updated the Metrics and Evaluation section (lines 415-423 in the revised manuscript) to make this aspect clearer.
>
> ---
>
> __4. the paper might be a better fit for the evaluations on dynamic knowledge graph completion.__
>
> Thank you for the suggestion. While our work is related to knowledge graph completion and temporal (dynamic) KG completion, our primary focus is different: we address information extraction from text and the continual updating of a KG based on newly emerging knowledge in textual sources. We have clarified this distinction in the revised manuscript (lines 139–149).

---

### Meta-Review · Area_Chair_Q3wx · 2026-01-04

**Summary:**

This paper proposes a new benchmark for updating knowledge graphs and demonstrates it using Wikipedia and Wikidata. This paper presents a method for building this KG structure. However, reviewers wRHi and pLRm point out several limitations of this approach and note that the proposed solution would not be beneficial for solving problems to which the KG could be applied. I strongly recommend that the authors consider the reviewer's suggestion before resubmitting to a machine learning conference. Also, a database or knowledge discovery conference might be a better venue for this work.

**Reviewer Concerns:**

The main concern of the reviewers were the lack of usefulness of the proposed benchmark and the low applicability of the generated KG.

**Reviewer Scores:**

The two negative reviewers (0 and 2 with high confidence) would not be convinced by the authors comments.

---

### Decision · Program_Chairs · 2026-01-26

Reject